# Nonlinear country-heterogenous impact of the Indian Ocean Dipole on global economies

Wenju Cai [1,2,3,4] ✉, Yi Liu [1,2], Xiaopei Lin [1,5], Ziguang Li[1,5], Ying Zhang [6] ✉ & David Newth[7]

A positive Indian Ocean Dipole features an anomalously high west-minus-east sea surface temperature gradient along the equatorial Indian Ocean, affecting global extreme weathers. Whether the associated impact spills over to global economies is unknown. Here, we develop a nonlinear and country-heterogenous econometric model, and find that a typical positive event causes a global economic loss that increases for further two years after an initial shock, inducing a global loss of hundreds of billion US dollars, disproportionally greater to the developing and emerging economies. The loss from the 2019 positive event amounted to US$558B, or 0.67% in global economic growth. Benefit from a negative dipole event is far smaller. Under a high-emission scenario, a projected intensification in Dipole amplitude causes a median additional loss of US$5.6 T at a 3% discount rate, but likely as large as US$24.5 T. The additional loss decreases by 64% under the target of the Paris Agreement.

A positive IOD (pIOD) develops in boreal summer (June-August) and matures in September, October and November[1,2] (SON), exerting a substantial impact. During the 1997 strong pIOD, floods[1,2] and malaria outbreak[3] occurred in East Africa, leading to fatalities in the thousands and displacement in hundreds of thousands. On the other side of the Indian Ocean, severe droughts and wildfires plagued Indonesia and Australia[2,4,5] and the associated smoke and haze affected the health and economic activities of tens of millions of people[6,7]. The impact extended to remote regions such as central and South America through atmosphere teleconnection[8–10]. The 2019 strong pIOD event saw similar effects. Sri Lanka, India and East Africa[11–13] experienced floods causing many deaths and displacing millions of people, whereas Indonesia and Australia saw a devastating bushfire season[14,15]. In eastern and southeastern Australia, the fires burned million hectares, destroyed 2400 buildings and killed 34 human lives[16] and over one billion animals[17]. The reduced output of Australia farming sector alone, known to be severely affected by the IOD (ref. [18,19]), amounted to about AU$5.0B (refs. [20]).

Studies of impacts from extreme weathers and climate have found ways for transmission to the broader economy[21–26]. For example, droughts and fires decreased employment in the farming sector and reduced demand; reduced farming production puts upward pressure on consumer prices[20]; reduced rainfall and surface run-off decrease economic activity of water-dependent industries[27]; poor weather conditions reduce tourist arrivals and consumptions for tourism-dependent regions[24]; smoke haze-induced air pollution from fires reduces workers productivity and increases road closures and uncertainty about safety[25]; and low crop yields and agricultural outputs lead to trade contractions and commodity price increases[26].

The potential spill overs have motivated studies of the impact on global economy, for example, from El Niño-Southern Oscillation (ENSO), the most consequential mode of climate variability originated in the tropical Pacific Ocean[28–32]. One type of econometric model assumes an impact symmetric about El Niño and La Niña but country-heterogenous governed by ENSO teleconnections[32]. Other models emphasize a nonlinear impact asymmetric about El Niño and La

[1]Frontiers Science Center for Deep Ocean Multispheres and Earth System/Physical Oceanography Laboratory/Sanya Oceanographic Institution, Ocean University of China, Qingdao, China. [2]CSIRO Environment, Hobart, TAS, Australia. [3]State Key Laboratory of Marine Environmental Science & College of Ocean and Earth Sciences, Xiamen University, Xiamen, China. [4]State Key Laboratory of Loess and Quaternary Geology, Institute of Earth Environment, Chinese Academy of Sciences, Xi'an, China. [5]Laoshan Laboratory, Qingdao, China. [6]School of Management, Ocean University of China, Qingdao, China. [7]CSIRO Environment, Black Mountain, Canberra, ACT, Australia. ✉ e-mail: Wenju.Cai@csiro.au; yzhang@ouc.edu.cn

Niña[29–31], because El Niño, stronger in amplitude than La Niña, typically results in 1–2% growth reduction to global economy[31,33], but the effect is virtually absent during La Niña[29–31]. These models converge on a persistent impact lasting for several years after the initial shock, causing a loss in trillions of US dollars during an extreme El Niño mostly through the spill-over and cascading effects[31]. Similar to ENSO asymmetry, a pIOD grows to a greater amplitude than that of an nIOD (ref. [4]); unlike ENSO, to date, the impact of the IOD on global economies is unknown.

Climate change affects global economy[34,35], and the associated impact is usually assessed using mean temperature and rainfall changes. The impact from changing climate variability under greenhouse warming is less known, in part because how climate variability itself might change is uncertain. Recent advances suggest that variability of the IOD is likely to increase over the 21st century under greenhouse warming, with moderate pIOD events being converted more frequently to strong pIOD events[15]. The extent to which the projected IOD change exacerbates the economic risks of greenhouse warming, particularly to the already vulnerable economies, is an important issue.

Here we establish a nonlinear econometric model that incorporates country-level impacts from the IOD on global economic growth. We find that a typical pIOD event leads to an economic loss of hundreds of billion US dollars, disproportionately large in developing economies, and that there is an exponential increase in the loss over the remaining 21st century from a projected intensification of future IOD.

## Results and discussion

### Nonlinear effect disproportionally large in developing economies

Unlike some econometric models that focus on different climate zones[29,30], here we incorporate a country-heterogenous assessment. For ENSO, because the impact on the global economy is strong and by and large worldwide, a country-heterogenous approach may not be essential[31]. By contrast, for the IOD, the impact is weaker and the affected regions are mostly developing economies. As such, the cascading effect and spillovers could be more regionally distinct, therefore a country-heterogenous approach is necessary.

Our country-heterogenous assessment is based on climate teleconnection of the IOD and population density, similar to previous approaches (see "Climate teleconnections" in "Methods"). We include the IOD climate teleconnections in fields of surface temperature and rainfall anomalies because both can simultaneously exert an impact. To combine their influence but avoid dominance by either anomaly field due to their amplitude contrast, the monthly anomaly field is normalized by its respective monthly standard deviation at individual grid-points. Influences from the IOD could commence and peak at different months in different regions, therefore we regress the normalized anomalies, from May to December covering the entire IOD life cycle of eight months, one month at a time, onto the IOD index[1] in SON, when an IOD peaks. Although most of strong pIODs such as the 1961, 1994, and 2019 events occurred independently of ENSO and many pIODs events occurred with a La Niña condition as in 2007 and 2008 (refs. [4,5]), some IOD events are accompanied by ENSO events. We therefore remove ENSO teleconnections through a partial regression, which appears to be effective (Supplementary Fig. S1).

For each field, we obtain eight monthly patterns of statistically significant regression coefficients; the regression coefficient values are then cumulated over the eight months (Supplementary Figs. S2, S3). The amplitudes of cumulated surface temperature and rainfall teleconnection are combined to form climate teleconnections and averaged over individual countries weighted by grid-point population density to yield a country-heterogenous IOD teleconnection pattern (Fig. 1a). Eastern African countries and Indonesia, both developing and

emerging economies, are among countries with the strongest teleconnections.

Because pIOD amplitude is greater than nIOD amplitude such that the impact may increase nonlinearly with the amplitude[4], and given that economic impact from climate variability can affect economic production in the ensuing years[29–31], our econometric model incorporates these features. As such, we build into a fixed-effect panel regression model[29–31,35] by adding the IOD teleconnections as a nonlinear predictor and incorporating lagged effects (see "Empirical econometric model" in "Methods"). The amplitude of the predictor is parameterized as the product of "IOD teleconnection and an IOD index". Our model is therefore continuously nonlinear and country-heterogenous, combining the characteristics of nonlinearity and country-heterogeneity.

We train our model using economic data and the IOD index over the 1960–2020 period. Like in the previous models[31,32] for ENSO, time-invariant and time-trending covariates are allowed to interact with observed economic and climate variables, and the nonlinear effect of mean temperature operates in addition to the IOD impact. We allow delayed impacts for a different number of years. A version with three years of delayed impacts (last column of Supplementary Table S1) shows that the IOD impact persists for another two years, after which time further impact is mild, as seen in the associated coefficients in year 3, when neither the linear nor the nonlinear is statistically significant. The impact is strongest in the year after the occurrence (year 1; year 0 is occurrence year) (Supplementary Fig. S4) when both the linear and nonlinear terms are statistically significant. In the occurrence year (year 0) or year 2 after the occurrence, only the linear term is statistically significant. Thus, the contemporaneous impact on economy is linear and mild but accelerates the year after, before tapering in year 2. This version is adopted. We focus on the effect for the three years (year 0 to year 2). A set of bootstrap methods are applied to quantify the uncertainty of point estimates in the model (see "Bootstrap tests" in Methods), and the tests show that the model is stable (Supplementary Table S2).

Our IOD impact model reveals a strong effect increasing exponentially with the IOD. For each IOD index value, we calculate the impact on Gross Domestic Production (GDP) growth cumulated over years 0, 1, and 2 for all countries, to establish the relationship between accumulated anomalous growth rate and the IOD index value. Firstly, there is an asymmetric impact between pIOD and nIOD, with the impact far greater than that for pIOD. For example, a 1.0 s.d. pIOD causes 1.89% (±0.64%) loss in Kenya GDP growth that is statistically significant, while a 1.0 s.d. nIOD causes 1.09% (±1.25%) benefit in Kenya GDP growth that is not statistically significant. Secondly, the nonlinear impact is greater for countries with a greater teleconnection (Fig. 1b). For example, the impact is greater in Kenya than in Australia; the difference is because the teleconnection is weaker in Australia than in Kenya, where weather-sensitive sectors are a substantial portion of a smaller economy.

For each IOD index value, the associated impact in terms of GDP values cumulated over years 0, 1, and 2 (in 2015 US dollars) is calculated by multiplying the IOD-induced change in GDP growth rate with the respective GDP value of the previous year, for all countries. During the 2019 pIOD, the total loss is approximately US$37.8B in Australia and US$4.7B in Kenya, larger in Australia because the Australia economy is larger. In some other developed countries, though with a weak teleconnection, the impact is similarly substantial because of a large economy, for example, the loss to US economy by the 2019 pIOD is US$62.3B (Fig. 1b).

The global impact is substantial. Aggregated over all countries, the 2019 pIOD event resulted in a loss of US$558B, amounting to a reduction in global GDP growth of 0.67% (Fig. 1c, d). More generally, for a one-standard deviation of a pIOD index value, there is an average loss in the global GDP growth rate of -0.24% (Fig. 1d). By contrast, the

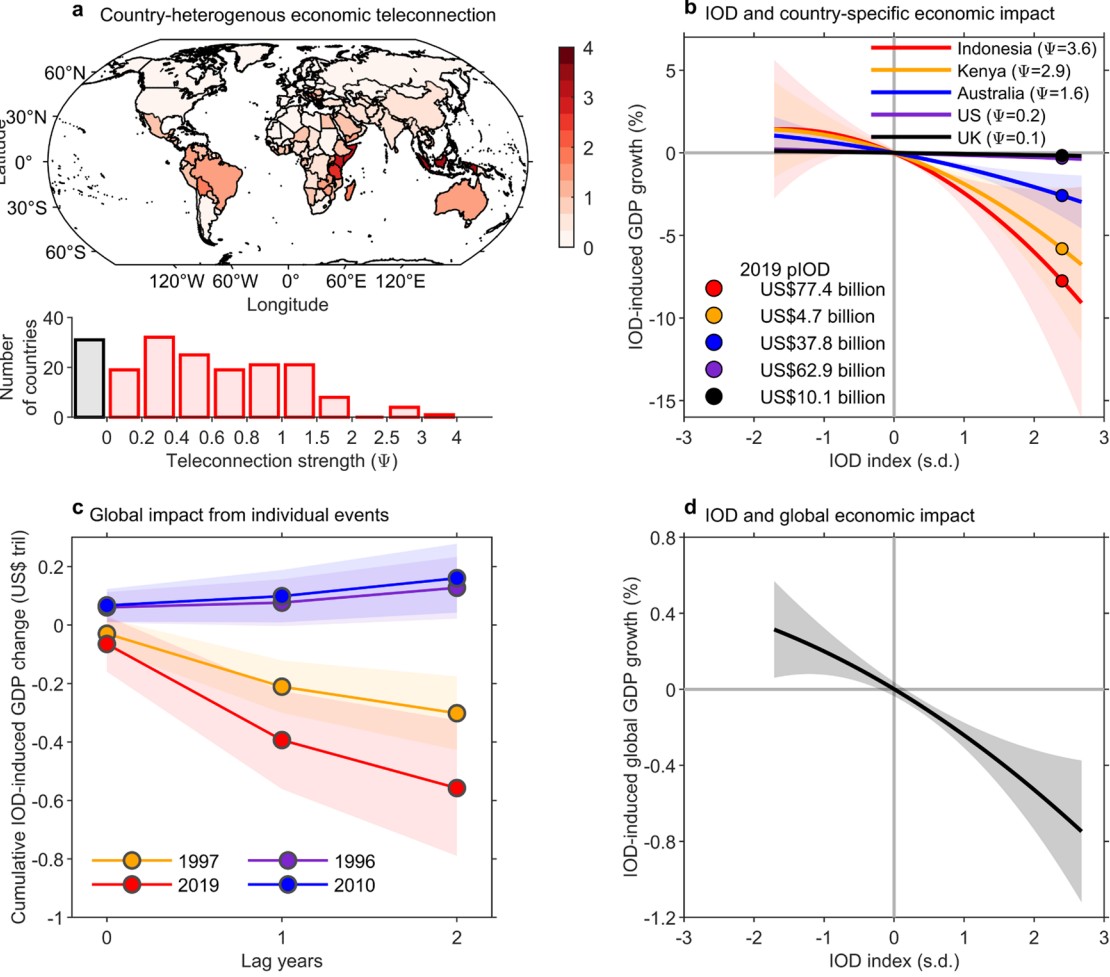

**Fig. 1 | Observed country-heterogenous economic impact. a** Global pattern of country-heterogenous teleconnection of the IOD, calculated as the sum of country-averaged, population-weighted climate teleconnection from the IOD in surface air temperature and rainfall (see "Climate teleconnections" in "Methods"). The bar chart below refers to the distribution of teleconnection, in which the red and black bar indicate the number of teleconnected and non-teleconnected countries, respectively. **b** Nonlinear relationships between the IOD index and IOD-induced impact on GDP growth for Indonesia (red), Kenya (yellow), and Australia (blue), with shadings indicating the 95% confidence interval based on a Bootstrap method

(see "Bootstrap tests" in "Methods"). Dots on each line represents values of the 2019 strong pIOD event, with estimated losses in GDP value shown in the legend. **c** Cumulative effect of two strong pIOD events in 1997 (yellow) and 2019 (red), and two strong nIOD events in 1996 (green) and 2010 (blue) on global GDP cumulating from year 0 (IOD occurrence year) to year 2. Shadings indicate the 95% confidence interval based on the Bootstrap method. **d** Same as (**b**) but for IOD-induced impact on global GDP growth. Impact of the pIOD and nIOD on global economic is asymmetric, far greater during pIOD events and increases nonlinearly with IOD amplitude, and disproportionally large on developing economies.

benefit from a nIOD event is hardly statistically significant (Fig. 1c, d), not only because of the smaller nIOD amplitude but also because of the nonlinear economic response. By comparison, there is a ~1.5% cumulative loss in global economy for one standard deviation of ENSO. The 2015/16 strong El Niño led to a loss of US$3.9 T, amounting to a ~5% cumulative loss in global economy[31].

**Asymmetric impacts render an economic loss over IOD cycles**
We calculate the effect of the IOD on economic production for a given year. Due to the lagged impact, for a given year $t$, the impact from the IOD on a country includes the contemporaneous and lagged effects from the previous two years. For example, the impact on GDP growth in 2019 includes the effect of year 2 of the 2017, year 1 of the 2018, and year 0 of the 2019 IOD index values. The evolution of the IOD impact on Kenya economic growth rate shows that pIOD events have largest negative impact in year 1 after occurrence and the associated economic loss is far greater than the economic benefit of an nIOD event (Fig. 2a). The timeseries of effect on GDP growth is negatively skewed, with a skewness of −1.74. The skewness of economic impact is additional to the IOD skewness, as reflected in a quadratic relationship

between the two time series after the IOD time series is shifted forward by one year to maximize their coherence due to the one year lag (Fig. 2b). The value of nonlinear coefficient, referred to as Ω, is −0.54.

As a result of the nonlinear economic impact, over time, the loss from pIOD is not offset by the benefit from nIOD. For Kenya, the net effect results in an averaged loss of 0.37% per annum averaged over the period, dominated by strong pIOD events. The amplitude of the nonlinear coefficient (Ω, Fig. 2b) in the relationship between temporal evolution of the loss and the IOD index measures the extent to which economic loss from pIOD outsizes the benefit of nIOD.

The value of nonlinear coefficient, computed for every country (Supplementary Fig. S5 for some other typical countries and global average) shows that across countries, the nonlinearity increases exponentially with the strength of the IOD teleconnection (Fig. 2c). In countries with a strong teleconnection, such as Kenya and Indonesia, the nonlinearity and hence the loss in growth rate is greater than in countries with a moderate teleconnection, such as Australia or Sri Lanka. Thus, the asymmetric impact between pIOD and nIOD renders an economic loss after a period of IOD cycles, and the loss increases exponentially with teleconnection strength.

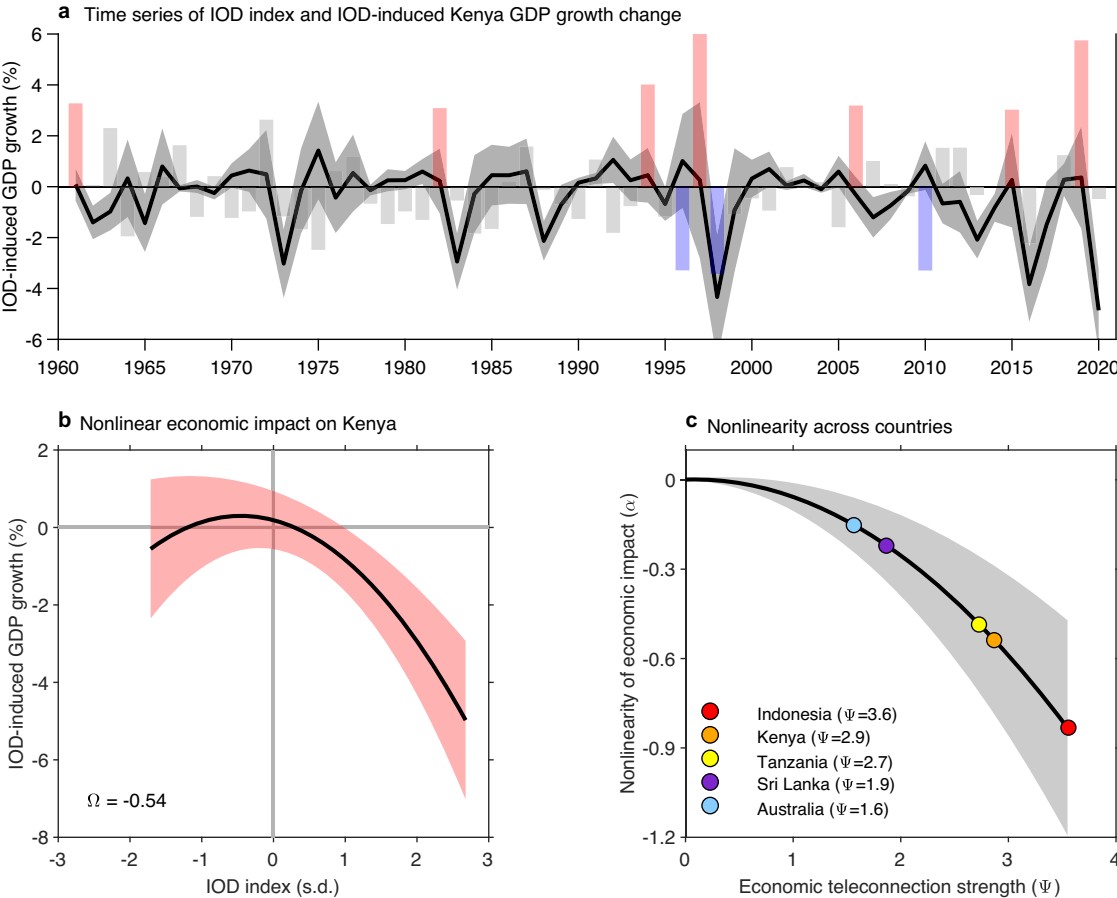

**Fig. 2 | Observed nonlinear economic impact of the IOD. a** Timeseries of Kenya's GDP growth rate change from the IOD (black line) and the IOD index (bars). Change in GDP growth rate is calculated as the cumulative contemporaneous effect of an IOD event in year 0 and growth effects of IOD events in year -1 and -2. Gray shading indicates the 95% confidence interval based on the Bootstrap method (see "Bootstrap tests" in "Methods"). Strong pIOD (SON IOD index >1.5 s.d) and nIOD (SON IOD index < −1.5 s.d) events are marked as red and blue bars, respectively.
**b** Nonlinear relationship between the IOD index and IOD-induced change in Kenya's GDP growth rate shifted backward by one year (both shown in Fig. 2a), with shading indicating the 95% confidence interval based on the Bootstrap method. A quadratic fitting, in which *GDP growth=constant + k×IOD + Ω×IOD²*, is performed and the coefficient for the nonlinear term Ω is shown. **c** Relationship between teleconnection and the nonlinearity Ω for all countries. Dots on the fitted line are examples of some strongly affected countries. Shading indicates the 95% confidence interval based on the Bootstrap method. Asymmetric impacts between pIOD and nIOD render an economic loss from IOD cycles, which increases exponentially with teleconnection strength.

We conduct a sensitivity test in which we include the ENSO's common global shock component[31] to our IOD economic impact model. The result from this combined model shows that the economic effect of the IOD is largely independent of that of ENSO. Although some of the effect is shared, the effect of the IOD in this combined model is only marginally smaller (19%) than that in our IOD-only economic impact model (Supplementary Fig. 6).

### Loss in economic growth increases with projected IOD variability

We apply our econometric model to examine the impact of changing IOD on global economy in a warming climate. We examine IOD changes under several emission scenarios of the Intergovernmental Panel on Climate Change (IPCC), using outputs of available climate models that participated in the Coupled Model Intercomparison Project phase 6 (CMIP6) (ref. 36). These models (Supplementary Table S3) are forced with historical anthropogenic and natural forcing until 2014, and four Shared Socioeconomic Pathways[37] (SSP) of future greenhouse gas concentration trajectories from 2015 onwards, that is, SSP5-8.5, SSP3-7.0, SSP2-4.5, and SSP1−2.6 (see "Climate scenarios, models and projection" in "Methods"). To describe model IOD evolution, an Empirical Orthogonal Function (EOF) analysis is applied to the detrended SST anomalies in the domain of 5°S-5°N, 40°E-100°E. The first EOF pattern,

which resembles the observed IOD pattern, and the associated principal component (PC) scaled to unity over the 200 years, are taken as our model IOD pattern and index. We use the ability of a model to simulate the IOD amplitude asymmetry between pIOD and nIOD to select models, as it is an indication of the realism of the model in simulating the IOD processes[15,38–40]. A total of 16 models are selected for comparison of the economic impact from changing IOD between different scenarios of greenhouse warming. We compare the standard deviation of the IOD index over the 2000-2099 (21st century) with that over the 1900–1999 period (20th century) to assess IOD changes.

There is a strong inter-model consensus on an increase in future IOD variability under all scenarios (Fig. 3a). In the SSP5-8.5, 87.5% of the models generate an increase in variability, with a multi-model ensemble median (mean) increase, of 29.5% (36.7%) translating to a higher frequency of strong pIOD events (Supplementary Fig. S8). For each model, IOD climate teleconnection is constructed in the same way as the observed with ENSO impacts removed. In terms of the teleconnection strengths, unlike the amplitude of IOD variability, multi-model ensemble mean shows no inter-model consensus nor a systematic pattern of a statistically significant change even under the high emission scenario (Supplementary Fig. S9). As such, we assess the economic impact of changes in IOD amplitude.

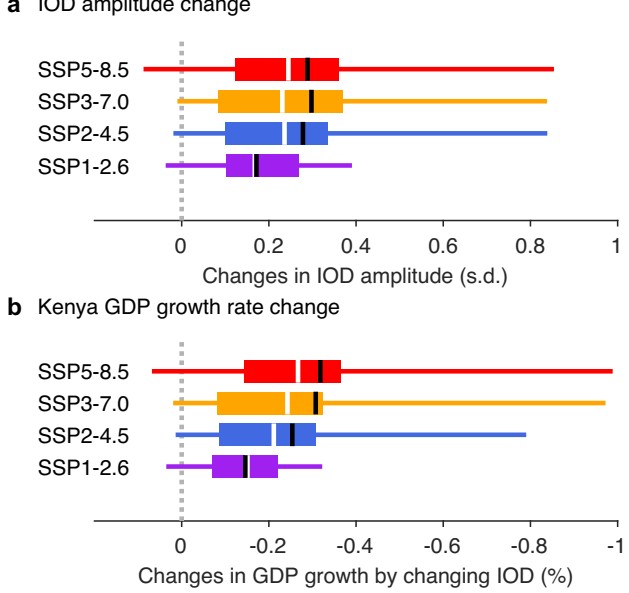

**a** IOD amplitude change

**b** Kenya GDP growth rate change

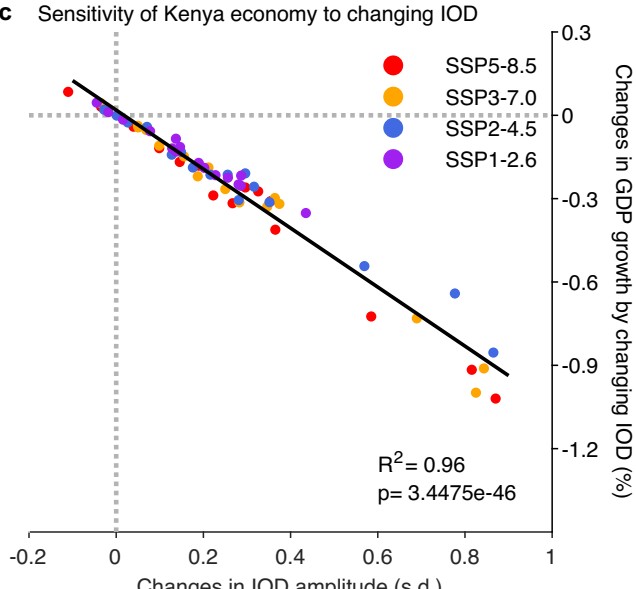

**c** Sensitivity of Kenya economy to changing IOD

**Fig. 3 | Projected changes in IOD amplitude and impact on Kenya economic growth. a** Changes in IOD amplitude from the 20th century (1900–1999) to 21st century (2000–2099) based on 16 CMIP6 models under four IPCC scenarios, with box indicating the range between the 25th and 75th percentile and whisker indicating the range between the 5th and 95th percentile. Colors refer to different IPCC scenarios. The multi-model ensemble median and mean for each scenario are shown as white and black vertical line, respectively. **b** Same as (**a**) but for changes in IOD-induced century-averaged GDP growth rate for Kenya, obtained as difference between results derived from the projected and counterfactual IOD time series over the 21st century, with the counterfactual having the same event sequences but the amplitude scaled to that over the 20th century. **c** Relationship between changes in IOD amplitude (21st century minus 20th century) and changes in Kenya GDP growth from the changing IOD amplitude. The R squared and p-value of a linear fit are given. Loss in economic growth increases with projected intensification of IOD variability in all emission scenarios.

We develop a counterfactual 21st century IOD time series such that it follows the same projected 21st century evolution and IOD event sequences but its standard deviation is scaled to the same amplitude as that in the 20th century (Supplementary Fig. S10). Sequences of IOD events can potentially affect the economic impact; because of the lagged effects, an IOD event, occurring early as opposed to at the end of the period, makes a difference. The counterfactual future IOD time series is taken as the IOD evolution if future IOD variability does not change. For each model and each country, using the projected and the counterfactual future IOD time series as input to our econometric model, we obtain a projected and a counterfactual time series of GDP growth change, where each value represents the combined effect of the contemporaneous and growth effects. The difference between them in GDP growth averaged over the 21st century is taken as a measure of the impact from the projected IOD change.

The reduction in economic growth increases with the projected IOD amplitude increase in all emission scenarios. Take Kenya as an example, an additional reduction is seen under all four IPCC emission scenarios, each supported by a strong inter-model consensus (Fig. 3b, c). The multi-model ensemble median (mean) increase in the century-average reduction of GDP growth is 0.27%, 0.24%, 0.21%, and 0.15% (0.32%, 0.31%, 0.25%, and 0.15%) per annum for the SSP5-8.5, SSP3-7.0, SSP2-4.5, and SSP1-2.6 scenario, respectively (Fig. 3b). Overall, there is a statistically significant sensitivity of an additional loss of 1.0% per annum per 1.0 s.d. increase in future IOD amplitude based on all CMIP6 models under the four emission scenarios (Fig. 3c). The inter-model consensus is seen in all affected countries, and the sensitivity increases with IOD teleconnection, stronger in Kenya and Indonesia than in Australia (Supplementary Fig. S11).

**Exponential increase in additional economic production loss from future IOD**
We estimate the IOD-induced economic production loss from the projected IOD based on the Shared Socioeconomic Pathways[37] (SSP)

that provide a secular evolution of country-level population and economic development. The SSP projections cover the period from 2010 to the end of 21st century, forced under the different emission scenarios. Assuming that losses from an IOD event permanently alter the long-term economic development, we construct two "no-IOD" future economic growth projections from 2020 to 2099, using the time series of growth rate from the projected and the counterfactual IOD time series, each including the contemporaneous and lagged effects, to construct the SSP economic growth rate time series. Two new GDP time series are obtained by compounding growth using the modified growth rate time series (Supplementary Fig. S10b). Additional GDP loss from changing IOD variability, discounted by different schemes ranging from 1 to 5% per annum, is calculated as the difference between the two time series of GDP. The change is then cumulated over the last 80 years (2020-2099). Such calculations are carried out for all countries and aggregated globally.

The projected increase in IOD variability likely causes substantial additional global economic losses. Using a 3% discount rate and bootstrapping the econometric regression model in 1000 different combinations of 181 countries allowing repeats, and 1000 realizations of IOD sequences from a given IOD time series (see "Sources of uncertainty" in "Methods"), the high emission scenario (SSP5-8.5) leads to a median additional global loss cumulative over the period of 2020-2099 is US$5.6 T (Fig. 4a). There is a large range in the projected loss with the 90% confidence interval spanning from a loss of US$24.5 T to a small benefit of US$0.5 T.

The same analysis is carried out for other emission scenarios. There is a more than 85% chance of an additional loss in any of the four emissions scenarios (Fig. 4b). Overall, the additional losses decrease with lowering emissions, due to commensurately smaller increases in the IOD amplitude. Under the SSP1-2.6 scenarios, a strong mitigation pathway to achieve the warming target of 1.5–2.0 °C warming relative to the pre-industrial levels, the smaller increase in IOD amplitude reduces the additional loss to a median US$2.0 T; though the loss is still

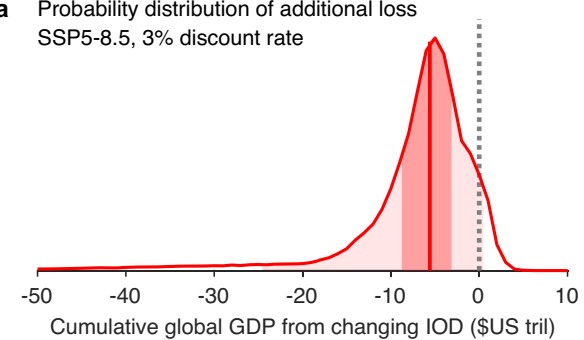

**a** Probability distribution of additional loss
SSP5-8.5, 3% discount rate

Cumulative global GDP from changing IOD ($US tril)

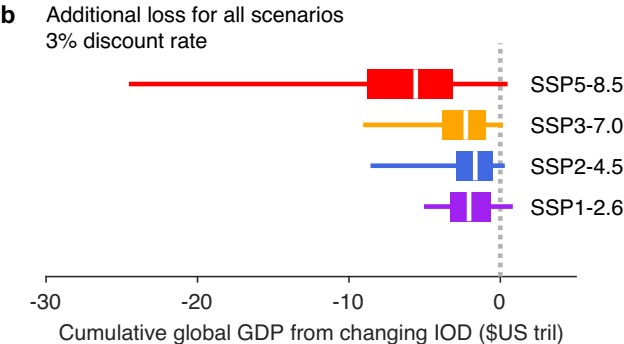

**b** Additional loss for all scenarios
3% discount rate

Cumulative global GDP from changing IOD ($US tril)

**c** Uncertainty from individual factors

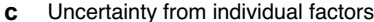
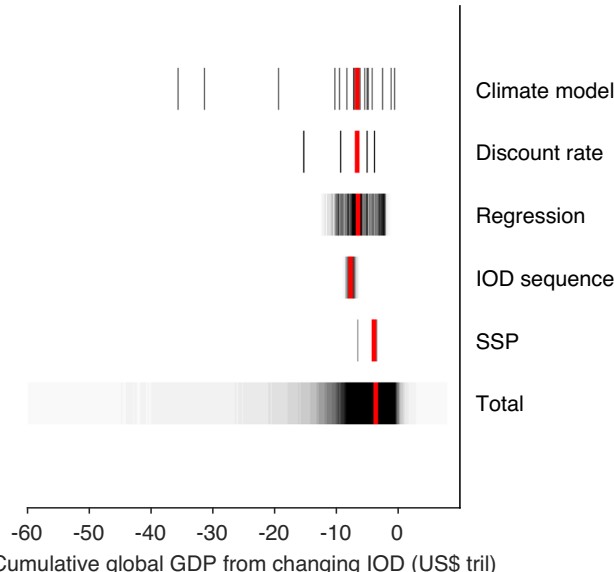

Cumulative global GDP from changing IOD (US$ tril)

**d** Nonlinear sensitivity to changing IOD

Fig. 4 | **Projected additional loss in global GDP from changing IOD. a** Probability distribution of estimated future change in global GDP resulting from changing IOD under SSP5-8.5 scenario at a 3% discount rate with uncertainty from econometric model regression and IOD sequences (see "Sources of uncertainty" in "Methods"), cumulative over the period of 2020–2099. Solid vertical line indicates the median value. Light and dark shading indicate the 5th-95th and 25th-75th percentile range, respectively. **b** Future changes in global GDP from changing IOD under four IPCC scenarios, with box indicating the range between the 25th and 75th percentile and whisker indicating the range between the 5th and 95th percentile range. White line shows the median value. **c** Sources of uncertainty including emission scenarios ('SSP'), different IOD amplitude changes in climate models ('Climate model'), choices of discount schemes ('Discount rate'), bootstrapped econometric model regression ('Regression'), and re-ordered sequences of IOD events ('IOD sequence'). The total uncertainty includes all these factors (see "Sources of uncertainty" in "Methods"). Each black vertical line is a point estimate (for example, there are four estimates for four SSP scenarios shown for the 'SSP' uncertainty), with red vertical line indicating the median value. **d** Nonlinear relationship between changes in IOD amplitude (21st century minus 20th century) and changes in global GDP from the changing IOD at a 3% discount rate cumulative over the period of 2020–2099, using the four IPCC scenarios. The R squared and p-value of a quadratic fit are given. Loss in economic production increases exponentially with projected intensification of IOD variability in all emission scenarios.

substantial, it represents a reduction of approximately 64% from the SSP5-8.5 scenarios, highlighting the benefit of mitigation.

The losses increase with a lower discount rate, or vice versa, which is one of the multiple sources of uncertainty, including model differences, emission scenarios, regression of the econometric model, and IOD event sequences. Assessment of their relative contribution to uncertainty, one by one by keeping other factors fixed at a chosen level (Fig. 4c), finds that the inter-model difference in IOD amplitude changes is the largest source of all factors. Discount rates are the second largest source whereas econometric model regression and sequences of IOD events contribute to only a small uncertainty range.

Importantly, in all scenarios, the additional loss in the 21st century increases exponentially with the projected increase in IOD amplitude (black line, Fig. 4d). The exponential increase is in turn underpinned by the nonlinear economic impact that is larger during pIOD

than during nIOD, in addition to the asymmetric IOD amplitude itself. The result reinforces the additional risk of increased IOD amplitude to global economy, with many models projecting an additional loss in the order of tens of trillion US dollars. On the flip side, the nonlinearity underpins the effectiveness of mitigation, as seen in the decreased loss in SSP1-2.6, in which curtailing the increase in IOD amplitude exponentially reduces the additional loss.

Alternatively, we calculate the global GDP growth rate with and without IOD changes for the four emission scenarios as in other studies[35,41], and then average the growth rate across all positive IOD events, defined as when an SON IOD index exceeds 1.0 s.d., over the 80-year period, for all emission scenarios (Supplementary Fig. S12). For example, the increased intensity of pIOD under SSP5-8.5 leads to an increase in loss from 0.20 to 0.29%, i.e., 45% greater because of the increased pIOD intensity alone, not counting the increased loss due to the increased frequency (Supplementary Fig. S13).

## Discussion

Our result of an increase in a substantial loss in global economy, disproportionally large in developing and emerging economies, from an increase in future IOD amplitude is underscored by an asymmetric impact that is greater during a pIOD than during a nIOD, and their two-year-long impact on economic growth that accelerates in the years after a pIOD occurrence. Because of these characteristics, there is a substantial net economic loss after a period of IOD cycles. Under a high emission scenario, increased IOD amplitude causes a median additional loss to the global economy that, over the last 80 years of the 21st century, cumulates to US$5.6 T but encompasses a potential loss as large as US$24.5 T under a 3% discount rate. The additional global economic losses increase exponentially with the projected increases in IOD amplitude. There is an incentive for achieving the warming target of the Paris Agreement as the associated smaller increase in IOD amplitude is projected to reduce the additional losses under the high-emission scenario by approximately 64%. Our finding of an unexpectedly large impact of the IOD on the global economy, and an exponential increase in economic loss highlights the increased risks, particularly to developing and emerging economies.

## Methods

### Climate teleconnections

We depict the observed IOD using the Ocean Reanalysis System 5 (ORAS5) over the period of 1958-2020 (ref. 42), a reanalysis that assimilates observed upper ocean temperatures and resolves key oceanic feedback responsible for the positive skewness of the IOD (refs. 38,39). Our IOD index is obtained from an empirical orthogonal function analysis[43] (EOF) on quadratically detrended SON SST anomalies in the equatorial Indian Ocean (5°S-5°N, 40°E-100°E), yielding a dipole pattern, and a principal component (PC1) that is scaled to unity; PC1 is highly correlated with the DMI with a coefficient r = 0.92. We use land-based surface air temperature and rainfall data from ECMWF Reanalysis version 5 (ERA5) (ref. 44) to construct the IOD climate teleconnection over the same period of 1958-2020.

The climate teleconnections In both surface temperature and rainfall anomalies are included, because both can simultaneously exert an impact; for example, during a pIOD event, lower rainfall and higher temperatures in Indonesia and Australia both contribute to drought and bushfires[4,5]. To combine their influence but avoid dominance by either anomaly field due to their amplitude contrast, monthly anomaly field is normalized by its respective monthly standard deviation at individual grid-point. Influences from the IOD could commence and peak at different months in different regions, we therefore examine climate teleconnection from May to December. We remove ENSO's influence on grid-point anomalies through a partial regression onto the IOD index (PC1) and an ENSO index (Niño3.4 SST) month by month.

Specifically, to remove impact from ENSO and to obtain IOD teleconnection for monthly surface air temperature $T_{x,y,m}$ at each longitude-latitude grid point $(x,y)$ and calendar month m from May to December, partial regression is calculated through[45]

$$\tau_{x,y,m} = \frac{R_{TI} - R_{TE} \cdot R_{EI}}{\sqrt{1 - R_{EI}^2}} \cdot \frac{\theta_T}{\theta_{I|E}} \tag{1}$$

where $\tau_{x,y,m}$ is the partial regression coefficient of $T_{x,y,m}$. $R_{TI}$ and $R_{TE}$ refer to correlation coefficients of $T_{x,y,m}$ with the IOD and with the ENSO index, respectively. $R_{EI}$ is the correlation coefficient between the IOD and the ENSO index. $\theta_T$ is the standard deviation of $T_{x,y,m}$ and $\theta_{I|E}$ is the standard deviation of the IOD index after linearly removing the ENSO influence. The same calculation is conducted for $P_{x,y,m}$ to obtain its partial regression coefficient $\rho_{x,y,m}$. The effectiveness of this approach is tested by applying to grid-point SST; a partial regression coefficient between grid-point SST anomalies after removing ENSO

and the IOD index shows no ENSO pattern in the equatorial eastern Pacific (Supplementary Fig. S1).

We add $\tau_{x,y,m}$ from May to December that are statistically significant above the 95% confidence level (Supplementary Fig. S2), to yield amplitude of the cumulative temperature teleconnection $\tau_{x,y}$. The same is carried out to $\rho_{x,y,m}$ to obtain amplitude of rainfall teleconnection $\rho_{x,y}$. The total "climate" teleconnection is the sum of temperature and rainfall teleconnection $\psi_{x,y} = \tau_{x,y} + \rho_{x,y}$ (Supplementary Fig. S3). The grid-point "climate" teleconnection is then converted to the country-heterogenous teleconnection $\psi_i$ by weighting the grid-point population density as of 2020 obtained from the Gridded Population of the World (GPW) version 4 (ref. 46), as shown in Fig. 1a.

### Empirical econometric model

We obtain country-level inflation-adjusted annual Gross Domestic Product (GDP) per capita from the World Bank Development Indicators[47], for all countries in the world over the period of 1960–2020, although data for only a subset of years are available for some countries. Data from Penn World Tables version 10.0 (ref. 48) is also used to test robustness of our results. We model the economic impact of the IOD on country-level GDP using a general distributed-lag regression model (ref. 49), in which country-heterogenous impacts of the IOD are incorporated through:

$$\log(y_{i,t}) - \log(y_{i,t-1}) = \sum_{l=0}^{n} \left\{ \alpha_{1,l} \psi_i I_{t-l} + \alpha_{2,l} (\psi_i I_{t-l})^2 \right.$$
$$\left. + \beta_{1,l} T_{i,t-l} + \beta_{2,l} T_{i,t-l}^2 + \lambda_{1,l} P_{i,t-l} + \lambda_{2,l} P_{i,t-l}^2 \right\} \tag{2}$$
$$+ \mu_i + \theta_{1,i} t + \theta_{2,i} t^2 + \varepsilon_{i,t}$$

where $y_{i,t}$ is GDP per capita in country $i$ and year $t$, $l$ is the lag year to year $t$, $I$ is the IOD index in year $t - l$, $\psi_i$ is the country-heterogenous teleconnection strength, $T$ and $P$ are annual mean surface air temperature and rainfall in year $t - l$, after removing the IOD's signal through linear regression.

The model includes a term $\mu_i$, which accounts for country-fixed, time-invariant factors, such as different history and culture backgrounds of individual countries. The model also includes country-level long-term linear and quadratic time trends in growth rates $\theta_{1,i} t + \theta_{2,i} t^2$ from changing political institutions and economic policies of individual countries, and nonlinear effects of annual mean country-level temperature and precipitation. The country-heterogenous impact of the IOD is a quadratic function of $(\psi_i I_{t-l})$ and allows lagged impacts accounting for growth effects after the initial shock contemporaneous with an IOD event. We have not included year-fixed effects, as they might weaken the statistical influence of the IOD, leading to an underestimation of the real impact[50]. Also, year-fixed effect could introduce risk of collinearity as IOD time-fixed effects could be correlated with a time-specific factor, making it hard to separate the impacts from the IOD.

We obtain the optimum duration of the IOD's growth effect by testing a different number of lagged years. The process finds that the IOD negatively affects global economy for another two years after the occurrence year, after which time a further increase is not statistically significant (Supplementary Table S1). During the occurrence year, the contemporary effect is mostly through a direct loss, which is mild in terms of a global total loss, likely because the climatic impact is largest in developing countries, in which the economies are relatively small. In the subsequent years in terms of climatic effect, drought and its effect such as drought-induced crop failures may continue but far weaker than in the occurrence year; however, the subsequent cascading effect becomes substantial due to impact on global trades, commodity prices, and low inventories, and due to a low capacity to deal with to the adverse effect, contributing to the nonlinear effect. As such, we

estimate the economic impact aggregated over the contemporaneous year and the two years after. The model generates a time series of the IOD-induced change in GDP growth.

Reducing the model to a nonlinear version without the country-heterogenous feature by setting the teleconnection to be a value of 1.0 in every country, we find a nonmeaningful result, in which both pIOD and nIOD could lead to a benefit in the occurrence year. The failure of such a model with a common global shock only is in contrast to the model of ENSO impact (ref. 31), in which a common global effect captures much of the impact. Including similar terms for ENSO, that is, nonlinear and country-heterogenous terms, generates coefficients that are not statistically significant, consistent with a previous finding that for ENSO a common global shock dominates[31]. Comparing to the impact of ENSO, impact from the IOD on global economy is about 16% of that from ENSO.

We construct an alternative model in which we add the ENSO's common global shock component[31] to our IOD impact model. The result from this combined model shows that the economic effect of the IOD is largely independent of that of ENSO. For example, the effect of the IOD from this combined model is only marginally smaller than that in the IOD-only economic impact model.

## Bootstrap tests

To quantify the uncertainty of estimated coefficients ($\alpha_1$ and $\alpha_2$) of econometric model, we use several of bootstrap[51] strategies. One is to sample by country, in which the 181 countries are randomly resampled to construct 1000 realizations of 181-element lists. A country is allowed to be selected again during the resampling. Then we use these resampled lists of countries to re-estimate Eq. (2), extracting the re-estimated coefficients $\alpha_1$ and $\alpha_2$. Another approach samples by year. The 60 years (1961-2020) are randomly resampled to construct 1000 realizations of 60-element lists, again allowing re-selection. The resampled lists of years are used to re-estimate Eq. (2) to obtained coefficients $\alpha_1$ and $\alpha_2$. In the third approach, we sample by five-year block. We group the data into five-year blocks (that is, 1961–1965, 1966–1970, …, 2016–2020), and resample these blocks as in other approach. The estimated coefficients are stable in all three strategies (Supplementary Tables S1, S2). We use strategy one in presenting our results.

We also use the Bootstrap method to test statistical significance of multi-model ensemble mean difference in IOD amplitude between the 20th and 21st century (Supplementary Fig. S7). The IOD amplitude in each century from each model are resampled randomly allowing repeat, to construct 1000 realizations of multi-model ensemble mean values. A multi-model ensemble mean difference greater than the sum of standard deviations of the 1000 realizations of the two centuries is statistically significant above the 95% confidence level.

## Climate scenarios, models and projection

**Scenarios.** To project future IOD change and the associated impact, we use climate outputs from climate models in the Coupled Model Intercomparison Project phase 6 (CMIP6) (ref. 36), and socio-economic outputs from Shared Socioeconomic Pathways (SSP) database version 2.0 (ref. 37). The climate models are forced with historical anthropogenic and natural forcing from 1850 to 2014, and different emission scenarios (SSP1-2.6, SSP2-4.5, SSP3-7.0 and SSP5-8.5, from low to high emissions) thereafter from 2015 to 2100. The first available experiment from each model is used (Supplementary Table S2). Monthly anomalies referenced to the climatology over the 1900–1999 period are constructed, and then quadratically detrended at each grid over the period of 1900-2099. The socio-economic projections contain different scenarios of socio-economic development, including country-level GDP and population, under various degrees of climate forcing from 2010 to 2100. The world GDP is projected to reach US \$1200 T, US\$330 T, US\$635 T, and US\$672 T by 2100 under SSP5, SSP3, SSP2, and SSP1.

**Model selection.** During strong pIOD, anomalous cooling in the east results as a consequence of nonlinear oceanic positive feedbacks along the equator, referred to as nonlinear oceanic zonal and vertical advection[38,39]. Anomalous cooling along the eastern equatorial Indian Ocean pushes atmosphere convection and the convergence zone to the west, generating strong equatorial easterlies, which in turn intensify the equatorial nonlinear oceanic zonal and vertical advection, conducive to the cooling[38,40]. The ability of a model to simulate the skewness is an indication of the realism of the model in simulating the IOD nonlinear oceanic positive feedbacks[15,38–40].

On this basis, SON grid-point SST anomalies referenced to the 1900–1999 climatology are constructed, an EOF analysis is applied to quadratically detrended SON SST anomalies in the domain of 5°S-5°N, 40°E-100°E. The first EOF pattern resembles the observed IOD pattern, and the associated principal component (PC1), scaled to unity over the 200 years, is taken as our model IOD index. The skewness of PC1 is used to select models. We use a threshold of 33% of the observed skewness, which is 0.7. Based on the high emission scenario of SSP5-8.5, we initially select 24 out of available 48 models using this threshold. However, outputs under other emission scenarios are available only in 16 common models. These 16 models are selected for assessment and comparison of economic impact from changing IOD under greenhouse warming.

**Projection.** We compare standard deviation of the IOD index over the 2000-2099 (21$^{st}$ century) with that over the 1900–1999 period (20$^{th}$ century) to assess changes in IOD variability and strong pIOD frequency under greenhouse warming in the selected models (Supplementary Fig. S6, S7). Under greenhouse warming, the eastern equatorial Indian Ocean warms slower than the west[4], facilitating a westward moment of atmospheric convection, and increased occurrences of strong pIOD events. As such, majority of models that simulate the positive skewness, that is, the positive feedbacks, produce an increase in IOD variability[15,38].

There is a strong inter-model consensus on an increase in future IOD variability. In the high-emission scenario of SSP5-8.5, a total of 14 out of 16 selected models (87.5%), but models under all scenarios generate a faster warming in the west than in the east[4], and a strong inter-model consensus on an increase in IOD variability, with a multi-model ensemble median (mean) increase of 29.5%, 29.2%, 26.2% and 19.1% (36.7%, 37.6%, 33.7%, 19.1%) for SSP5-8.5, SSP3-7.0, SSP2-4.5, and SSP1-2.6, respectively (Fig. 3a; Supplementary Fig. S7). The increase in IOD variability translates to a higher frequency of strong pIOD events, defined as when the IOD index is greater than a 1.5 standard deviation value (Supplementary Fig. S8).

Combined climate teleconnection from surface temperature and rainfall is constructed in the same way as for the observed and shows no systematic change (Supplementary Fig. S9), in contrast to the IOD amplitude. We therefore examine the impact of the changing IOD amplitude, assuming that the econometric model based of historical climate and economic data operates.

## Sources of uncertainty

Our projection of additional GDP loss in is affected by uncertainty in several factors including projected socio-economic baseline, emission level, discount schemes, econometric model regression, IOD amplitude changes, and sequence of IOD events (five factors). We test their relative influence one by one. For example, to quantify uncertainty associated with IOD event sequences, we re-order an IOD timeseries to generate 1000 lists; for each list we re-build the counterfactual IOD timeseries and GDP growth to estimate the future additional loss. To do so, we hold four out of five factors fixed: SSP scenario is fixed to SSP5-8.5, climate model projection of IOD amplitude changes is fixed to the multi-model ensemble median value, discount rate is fixed at 3%, econometric model regression is fixed to the original point estimate,

and IOD sequence is fixed to the original sequence from each climate model. The total uncertainty is a combination of the above sources of uncertainty, in which each factor is allowed to change.

The total uncertainty encompasses a 95% confidence interval of changes in global economic loss from -US$22.5 trillion to US$0.7 trillion, while SSP scenario uncertainty alone leads to a 95% confidence interval of -US$5.0 trillion to -US$1.9 trillion, climate model uncertainty to a 95% confidence interval of -US$33.6 trillion to US$1.0 trillion, discount rate uncertainty to a 95% confidence interval of -US$13.9 trillion to -US$2.2 trillion, econometric model regression uncertainty to a 95% confidence interval of -US$9.4 trillion to -US$0.6 trillion, and IOD sequence uncertainty to a 95% confidence interval of -US$7.1 trillion to -US$5.1 trillion (Fig. 4c). We highlight that climate models with varying IOD amplitude changes contribute most to the total uncertainty, reinforcing that the additional GDP loss is underpinned by IOD amplitude change (Fig. 4d).

## Data availability
All datasets related to this paper are publicly available and can be downloaded from the following websites:

ORAS5: https://www.ecmwf.int/en/forecasts/dataset/ocean-reanalysis-system-5

ERA5: https://www.ecmwf.int/en/forecasts/dataset/ecmwf-reanalysis-v5

ERSSTv5: https://psl.noaa.gov/data/gridded/data.noaa.ersst.v5.html

World Bank Development Indicators: https://databank.worldbank.org/source/world-development-indicators

Penn World Tables v10.0: https://www.rug.nl/ggdc/productivity/pwt/

GPWv4.11: https://sedac.ciesin.columbia.edu/data/set/gpw-v4-population-density-adjusted-to-2015-unwpp-country-totals-rev11

CMIP6: https://esgf-node.llnl.gov/search/cmip6/

SSP database version 2.0: https://tntcat.iiasa.ac.at/SspDb/dsd?Action=htmlpage&page=10.

## Code availability
Codes for generating all the results are available from the corresponding authors on request.

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

## Acknowledgements

This study is supported by the National Science Foundation of China (42176218, Y.Z) and the Strategic Priority Research Program of the Chinese Academy of Sciences (XDB40030000). X.L. and Z.L. are supported by the National Natural Science Foundation of China (41925025 and 92058203). Y.L. is supported by the Fundamental Research Funds for the Central Universities (202261003) and the China Scholarship Council (202106330019).

## Author contributions

W.C. and Y.L. conceived this study and wrote the initial manuscript. Y.L. performed the analyses in discussion with W.C. and Y.Z. X.L., Z.L. and D.N. contributed to interpreting results, discussion and improvement of this paper.

## Competing interests

The authors declare no competing interests.
