## [Peer Review File · Nature Communications]

Nonlinear country-heterogenous impact of the Indian Ocean Dipole on global economiesReviewers' Comments:

Reviewer #1:

Remarks to the Author:

Report for NCOMMS-23-47036

Nonlinear country-specific impact of the Indian Ocean Dipole on global economies

The article aims to measure the economic effects generated by the IOD phenomenon on 180 countries using an ARDL econometric model and a country-specific method based on a teleconnection strength parameter. This article is closely aligned in philosophy and methodology with a study published in 2023 in Nature Communications on the individualized economic effects of ENSO. It is well-structured, the empirical work appears well-executed, and it holds interest for publication in a journal like Nature Communications.

However, I would like to modestly address some points through questions and requests for clarification that could further enhance the article in the context of a revision.

- A significant contribution of the paper lies in highlighting non-linear effects of IOD on production. However, could this result stem from the chosen modeling, which might overestimate the presence of non-linear effects by using the quadratic form ($\alpha_1 + \alpha_2$)? Have you tested a linear model and compared its predictive capabilities to your non-linear model? If the non-linear model exhibits superior predictive abilities, then it is legitimate to use and estimate it; otherwise, doubts might arise about the factors driving the results.
- Have you tested a common IOD shock in addition to interaction terms, similar to your previous article on ENSO, which is methodologically very close?
- Do T and P capture the residual effects of temperatures and precipitation after controlling for IOD (and also ENSO) effects, is that correct? Could there also be an indirect effect captured by an interaction term like $T \cdot IOD$, for example?
- You mention using a 2-lag model, deemed optimal, but based on what criteria? The S1 table shows that lag3 has a higher R-squared. Of course, I am not sure if it is the raw or adjusted R-squared; I presume it is the adjusted R-squared.
- Only the α_{2_1} coefficients are significant at 1%, but the rest are significant at 10% only. Isn't this problematic given the large number of observations you have?
- The effects of the squared term are significant, but not for lag0 (contemporary effect), and similarly, linear effects are non-significant for lag0 and lag1. How do you explain this from both a climatic and economic perspective? What happens in terms of channels/mechanisms to explain this?
- The article controls for several sources of uncertainty, which is very interesting. However, a potential source of bias and uncertainty is not mentioned, in my opinion: the quality and relevance of GDP data in certain countries, especially in developing and African countries, where data quality and availability may be low. What about that? (the authors used both World Bank and Penn WT but what about Maddison or other data). This could potentially affect the dynamics in ARDL models in time series and the significance of lags. Would it not be interesting to test the model's robustness since 1980 or 1990 as a robustness exercise?
- We observe, especially with Figure 1a, that the vast majority of countries have a teleconnection strength coefficient below 1.5, and unlike ENSO, as the authors state, the distribution is more asymmetric. Perhaps this justifies the non-linear model. However, it would be interesting to know the economic damages recorded by moderately teleconnected or even weakly teleconnected countries and

to add them to Figure 1.b, which only concerns three of the most teleconnected countries. In terms of economic policy impact and for the general public, knowing that IOD could have significant effects of a certain magnitude in the United States or European countries could have a significant impact on the fight against climate change, as one might expect less damage from this type of teleconnections.

- In Figure 4b, is it logical for SSP2-4.5 to be associated with a negative coefficient of such a small magnitude?

- It would be interesting in the discussion to provide a comparison with ENSO results to put in perspectives the magnitude of losses caused by IOD at a global and regional levels.

- Regarding climate variability and climate change effects on the global economy, it might be good to cite, in addition to references 31 and 32, the two articles by Kotz et al. (2021, 2022) in Nature Climate Change and Nature because they are recent significant contributions on this topic and address climate variability for temperatures and precipitation.

- The article's form could be improved. There are a few typos, such as 'Rsquare' instead of 'RSquared' as an example.

Reviewer #2:

Remarks to the Author:

The authors provide a timely and interesting analysis of the impacts of the Indian Ocean Dipole on the economic growth of countries across the world. The paper builds on standard climate-econometric techniques but applies a nuanced understanding of climate science including EOF analysis and assessments of teleconnection strength to construct a powerful indicator of national-level exposure to IOD. The empirical results are interesting and the combination with future projections of IOD make for highly relevant findings.

However, I have a number of questions regarding the methodology and framing which, for me, would be essential to address before publication. I list these in order of importance:

Major:

1. The independence of impacts from ENSO

Recent literature, including by the authors, has shown strong impacts of El Nino on economic growth. While the present analysis takes care to define the IOD teleconnections as independent of El Nino, they do not assess whether the economic impacts from IOD which they identify are independent from those caused by El Nino. Given that we know that positive IOD are often associated with El Nino, it is crucial for our understanding and the implications of the results to assess an empirical model which includes both an El Nino and an IOD index as independent variables to test how much they independently impact global economic output.

2. Projections of future impacts

The calculation of future economic impacts due to the IOD change should be made clearer, currently there are no equations for this in the methods section. In particular, the description on L241-246 was unclear to me in how the discount rate was applied. The discount rate should be used to weight the cumulative sum of future GDP losses, but in these lines it is described as being applied to the compounding growth rates. This method needs to be clarified before I can assess its validity.

Moreover, presenting economic damages as cumulative GDP losses over the whole of the century likely

inflates the numbers compared to other presentations. Most assessments of future economic impacts of climate change typically report impacts as an annual % reduction of future GDP. Obviously, the authors are at liberty to choose how they present their results, but I think it would be valuable and make for a more honest comparison to at least also present the impacts of IOD that they calculate in this way which is consistent with other literature (e.g. Burke 2015, Kalkuhl & Wenz 2020).

3. Bootstrapping and significance of empirical regressions

The authors should include additional tables which show the results of the assessment of statistical significance when using the other two bootstrap resampling approaches. Moreover, the authors should discuss what assumptions and types of uncertainty these different bootstrapping approaches are likely to assess and justify their main choice. I think this is particularly important to consider in light of the fact that the main independent variable (IOD) is common, and therefore perfectly correlated, across all countries. This means there are much fewer independent observations than normally assumed in a panel regression where independent variables are not so correlated across groups. This issue compared to normal panel models should be explicitly dealt with when presenting the significance of the econometric results.

4. Climate teleconnection

This is an interesting and subtle way to construct a countries potential exposure to IOD. However, a few methodological choices seem subjective and alternative approaches are not explored. E.g.:

Can the authors demonstrate the effectiveness of the partial regression by also showing the direct correlation of sea surface temperatures with IOD without removing the ENSO effect?

Why are climate teleconnections only from May to December included?

Minor:

5. Fixed-effects choices

Panel fixed-effects models typically include two way (country and year) fixed effects for causal identification. The year fixed effects capture unobserved confounders which are common across countries in given years. Here, these are not included. This is likely due to the fact that IOD variation is common across countries and so including year-fixed effects would remove the variance in which the model is interested in identifying. So this is a justifiable choice, but the authors should make that justification explicit.

6. Description as "country-specific" results

In my opinion, the description of the empirical model as "country-specific" is mis-leading. A global panel model estimates an aggregated response across countries. Even when including interaction terms (here with the teleconnection strength) to describe cross-country heterogeneity, the model still estimates a global heterogeneity based on this strength rather than a specific country-level response.

L144 – do the authors distinguish between countries with a larger/smaller agricultural share in their econometric specification? It appears not based on their methods, and that the only way different impacts across countries are described in the model are through the teleconnection strength. Therefore it seems like this interpretation of their results that Australia suffers a larger impact due to its larger agricultural sector is not justified by their methods or results.

L133 – can the authors show a table indicating this insignificant result?

140 - the impact is of an opposite sign for negative/positive IOD and this should be stated, not just that the impact is greater for positive than negative.

L207-210 – should be made clearer that these sentences refer to the teleconnection strengths/patterns.

There are a number of minor wording issues which should be corrected, in particular missing “the”, e.g. L87 “the impact on the global economy”.

Reviewer #3:
Remarks to the Author:
Please see attachment.

Referee report for “Nonlinear country-specific impact of the Indian Ocean Dipole on global economies”

This paper estimates the effect of the Indian Ocean Dipole (IOD) on economic growth, and projects the economic impacts of an intensified IOD amplitude under future climate change. The IOD is a mode of interannual global climate variability originating in sea surface temperature anomalies in the equatorial Indian Ocean. These anomalies are correlated with strong variation in surface air temperature and rainfall in some parts of the world (i.e., highly “teleconnected”) but much less so in other parts of the world (i.e., not highly “teleconnected”). The paper exploits these differences in teleconnection to estimate the effect of the IOD on economic growth by country. An important finding is that the effects are highly nonlinear/asymmetric- while a large positive IOD has a large negative effect on economic growth, a negative IOD has virtually no effect on growth. The harmful effects of positive IODs are stronger for more teleconnected countries. Based on these findings, the paper projects that intensified IOD amplitude under climate change will have large economic costs compared to a no-climate-change counterfactual. These costs will be disproportionately borne by today’s developing and emerging economies, as these happen to be the ones with greatest teleconnection. (Climate change is not projected to alter the patterns of teleconnection.)

The basic research design of this paper combines previous work on the effects of another mode of interannual global climate variability- ENSO (e.g., Hsiang et al., 2011) with previous work on the effects of climate on economic growth (e.g. Burke et al., 2015). ENSO originates in sea surface temperature anomalies in the eastern equatorial Pacific and its effects have been studied considerably more than those of the IOD. As such, this paper fills in important gap by studying the economic effects of the IOD.

While the paper takes on an important topic, more work needs to be done on the econometric estimation and climate change impact projections to ensure the robustness of the results. The main estimating equation (Equation 2) includes a multiplicative interaction of an annual IOD index value with a country-specific measure of teleconnection. This interaction term is modeled as a quadratic function to account for nonlinearity in the country-specific effects of IOD. Additional regressors include country-level annual mean surface air temperature and rainfall.

1. Importantly, the IOD index values used in the estimating equation are constructed via a partial regression that removes the impact of ENSO (Equation 1). This approach seems opaque. The authors should consider also running an alternative version of the estimating equation that includes as regressors both the IOD index x IOD teleconnection interaction as well as an ENSO index x ENSO teleconnection interaction. Such a specification will directly control for the effect of ENSO and will allow for a transparent comparison between the effects of IOD and ENSO. Both effects could then also be carried through to the climate change impact projections, allowing for a comparison of the respective impacts.
2. The estimating equation includes country-level annual mean surface air temperature and rainfall, both modeled as quadratic. However, the estimated effects of these variables on economic growth are never displayed, nor do they seem to be accounted for in the climate change impact projections. Considering that prior work has shown temperature (Burke et al., 2015) and rainfall (Kotz et al., 2022) to affect economic growth, the authors should present the estimated effects of these variables and carry these forward when projecting the impacts of climate change. This will allow for comparison between the impacts of different climatic variables on future economic costs.

3. The estimating equation includes contemporaneous and lagged effects of all variables. However, only the cumulated effect is ever plotted. It would be useful to also see the separated effects presented in a supplementary figure.

Other Points

1. Figure 2c plots the nonlinearity of economic impact (referred to as alpha) against economic teleconnection strength. It is unclear to me what exactly is alpha as I could not find it defined anywhere. The estimating equation (Equation 2) refers to alpha_1 and alpha_2, but not alpha without a subscript.
2. There are multiple references to an acronym “SON”, but I could not find it defined anywhere.

References

Burke, M., Hsiang, S.M. and Miguel, E., 2015. Global non-linear effect of temperature on economic production. *Nature*, 527(7577), pp.235-239.

Hsiang, S.M., Meng, K.C. and Cane, M.A., 2011. Civil conflicts are associated with the global climate. *Nature*, 476(7361), pp.438-441.

Kotz, M., Levermann, A. and Wenz, L., 2022. The effect of rainfall changes on economic production. *Nature*, 601(7892), pp.223-227.

We thank three reviewers for the helpful and positive comments. Please find our response below (in blue).

Response to Reviewer #1

The article aims to measure the economic effects generated by the IOD phenomenon on 180 countries using an ARDL econometric model and a country-specific method based on a teleconnection strength parameter. This article is closely aligned in philosophy and methodology with a study published in 2023 in Nature Communications on the individualized economic effects of ENSO. It is well-structured, the empirical work appears well-executed, and it holds interest for publication in a journal like Nature Communications. However, I would like to modestly address some points through questions and requests for clarification that could further enhance the article in the context of a revision.

Thank you very much for your detailed, thoughtful, and helpful comments.

- A significant contribution of the paper lies in highlighting non-linear effects of IOD on production. However, could this result stem from the chosen modelling, which might overestimate the presence of non-linear effects by using the quadratic form ($\alpha_1 + \alpha_2$)? Have you tested a linear model and compared its predictive capabilities to your non-linear model? If the non-linear model exhibits superior predictive abilities, then it is legitimate to use and estimate it; otherwise, doubts might arise about the factors driving the results.

Thank you for the helpful comments. We did test using a linear model. The resultant Alpha coefficients are similarly statistically significant in the two years after the occurrence (see **Table 1-R1**).

Nonlinear model				Linear Model	
$\alpha_{1,0}$	-0.0015* (8.51×10^{-4})	$\alpha_{2,0}$	3.43×10^{-4} (2.60×10^{-4})	$\alpha_{1,0}$	-9.39×10^{-4} (7.66×10^{-4})
$\alpha_{1,1}$	-0.0018* (9.01×10^{-4})	$\alpha_{2,1}$	-7.52×10^{-4} ^{4***} (2.62×10^{-4})	$\alpha_{1,1}$	-0.0029*** (8.09×10^{-4})
$\alpha_{1,2}$	-0.0017* (9.00×10^{-4})	$\alpha_{2,2}$	-4.99×10^{-5} (2.82×10^{-4})	$\alpha_{1,2}$	-0.0014* (8.47×10^{-4})
$\alpha_{1,3}$	-7.15×10^{-4} (9.03×10^{-4})	$\alpha_{2,3}$	-2.51×10^{-4} (2.79×10^{-4})	$\alpha_{1,3}$	-0.0011 (8.26×10^{-4})

Table 1-R1 | Comparison of linear and nonlinear models.

We use a nonlinear model because **either an extreme positive or an extreme negative IOD causes a damage**, which would not be captured by a linear model. The superior performance of the nonlinear model is supported by the fact that it explains ~50% more variability than that explained by the linear model; for example, the nonlinear model shows a standard deviation of 1.29% of variability in Kenya GDP growth rate over the 60-year period, but the value for the linear model is only 0.84%.

- Have you tested a common IOD shock in addition to interaction terms, similar to your previous article on ENSO, which is methodologically very close?

We did. If we use the same common shock model as in our ENSO model, we obtain a nonmeaningful result. For example, in the occurrence year the impact is a benefit for both

positive IOD and negative IOD events. For the IOD, a country-specific resolution is essential. We have included this information in the revised version. See Lines 384-387.

- Do T and P capture the residual effects of temperatures and precipitation after controlling for IOD (and also ENSO) effects, is that correct? Could there also be an indirect effect captured by an interaction term like T*IOD, for example?

Our model adds the IOD impact to the original model – and this approach captures the residual effect after controlling for T and P that are independent from the IOD, that is, after the influence by the IOD on T and P is removed. Your idea of an interaction term like “*T*IOD*” is a worthwhile one. It could be a way to study how warming might exacerbate the impact of the IOD. We have tried to introduce such a term (see the $(\gamma_l T_{i,t-l} I_{t-l})$ term below) but have not been able to find coefficients that are statistically significant (**Table 2-R1**). Whether this is anything to do with our short data length is unclear, and is definitely a future direction to pursue.

$$\log(y_{i,t}) - \log(y_{i,t-1}) = \sum_{l=0}^n \{ \alpha_{1,l} \psi_l I_{t-l} + \alpha_{2,l} (\psi_l I_{t-l})^2 + \beta_{1,l} T_{i,t-l} + \beta_{2,l} T_{i,t-l}^2 + \gamma_l T_{i,t-l} I_{t-l} + \lambda_{1,l} P_{i,t-l} + \lambda_{2,l} P_{i,t-l}^2 \} + \mu_i + \theta_{1,i} t + \theta_{2,i} t^2 + \varepsilon_{i,t}$$

γ_0	-1.21×10^{-4} (6.17×10^{-4})
γ_1	-2.29×10^{-4} (6.45×10^{-4})
γ_2	-3.88×10^{-5} (6.71×10^{-5})
γ_3	-8.56×10^{-5} (6.59×10^{-5})

Table 2-R1 | Coefficients for interaction term, not statistically significant at any lags.

- You mention using a 2-lag model, deemed optimal, but based on what criteria? The S1 table shows that lag3 has a higher R-squared. Of course, I am not sure if it is the raw or adjusted R-squared; I presume it is the adjusted R-squared.

It is adjusted R-squared.

- Only the alpha2_1 coefficients are significant at 1%, but the rest are significant at 10% only. Isn't this problematic given the large number of observations you have?

The 10% significance is definitely not high but is unusual, and certainly what we can get. The lag 2 Alpha2 at the 1% significance suggests that the spillover or cascading effect is largest the year after the occurrence. This appears reasonable as the cascading effect takes time to be reflected in the economic data the year after. It is not clear how the result might be different if we would have more data than 60 years.

- The effects of the squared term are significant, but not for lag0 (contemporary effect), and similarly, linear effects are non-significant for lag0 and lag1. How do you explain this from both a climatic and economic perspective? What happens in terms of channels/mechanisms to

explain this?

Great question. During the occurrence year, the contemporary effect is mostly through a direct loss, which is mild, in part likely because the climatic impact is largest in developing countries, in which the economies are relatively small. In the subsequent years drought and its effect such as drought-induced crop failures may continue but weaker than the occurrence year; however, the cascading effect becomes substantial due to impact on global trades, commodity prices, and low inventories, and from a low capacity to deal with the adverse effect, contributing to the nonlinear effect. We have added this information in the revised version. See Lines 375-381.

- The article controls for several sources of uncertainty, which is very interesting. However, a potential source of bias and uncertainty is not mentioned, in my opinion: the quality and relevance of GDP data in certain countries, especially in developing and African countries, where data quality and availability may be low. What about that? (the authors used both World Bank and Penn WT but what about Maddison or other data). This could potentially affect the dynamics in ARDL models in time series and the significance of lags. Would it not be interesting to test the model's robustness since 1980 or 1990 as a robustness exercise?

Great suggestions. Our robustness test is mainly through the Bootstrap method. We have added a Supplementary Table (S2) regarding robustness with different years used.

We use data that are commonly used, for example, by Burk et al. 2015, Kotz et al. (2021, 2022), and Callahan et al. (2023). When we fit the model with the data since 1990 or 1980, some of the originally significant coefficients become not significant due to a shorter length of data (**Table 3-R1**). Length of data seems rather important.

1960-2020				1990-2020			
$\alpha_{1,0}$	-0.0015* (8.51×10 ⁻⁴)	$\alpha_{2,0}$	3.43×10 ⁻⁴ (2.60×10 ⁻⁴)	$\alpha_{1,0}$	-8.96×10 ⁻⁴ (0.0011)	$\alpha_{2,0}$	6.38×10 ⁻⁴ (3.04×10 ⁻⁴)
$\alpha_{1,1}$	-0.0018* (9.01×10 ⁻⁴)	$\alpha_{2,1}$	-7.52×10⁻⁴*** (2.62×10⁻⁴)	$\alpha_{1,1}$	-0.0015 (0.0012)	$\alpha_{2,1}$	-6.94×10⁻⁴** (3.10×10⁻⁴)
$\alpha_{1,2}$	-0.0017* (9.00×10 ⁻⁴)	$\alpha_{2,2}$	-4.99×10 ⁻⁵ (2.82×10 ⁻⁴)	$\alpha_{1,2}$	-0.0014 (0.0012)	$\alpha_{2,2}$	-1.16×10 ⁻⁴ (3.46×10 ⁻⁴)
$\alpha_{1,3}$	-7.15×10 ⁻⁴ (9.03×10 ⁻⁴)	$\alpha_{2,3}$	-2.51×10 ⁻⁴ (2.79×10 ⁻⁴)	$\alpha_{1,3}$	-3.29×10 ⁻⁴ (0.0012)	$\alpha_{2,3}$	-1.61×10 ⁻⁴ (3.41×10 ⁻⁴)

Table 3-R1 | Coefficients for using data since 1990.

The Maddison data appear to be of lower quality and not used by research on a similar topic.

- We observe, especially with Figure 1a, that the vast majority of countries have a teleconnection strength coefficient below 1.5, and unlike ENSO, as the authors state, the distribution is more asymmetric. Perhaps this justifies the non-linear model. However, it would be interesting to know the economic damages recorded by moderately teleconnected or even weakly teleconnected countries and to add them to Figure 1.b, which only concerns three of the most teleconnected countries. In terms of economic policy impact and for the general public, knowing that IOD could have significant effects of a certain magnitude in the United States or European countries could have a significant impact on the fight against climate change, as one might expect less damage from this type of teleconnections.

Thanks for the suggestion; see the result below for USA, for example, in terms of percentage of economic growth, it is small, but the total loss is ~60 billion of US dollars for the 2019 pIOD event (**Fig. 1-R1**). We have added this information in the paper, see Lines 153-155.

Fig. 1-R1 | Same as Fig. 1b, but adding the nonlinear relationship of United States (green) and United Kingdom (black).

- In Figure 4b, is it logical for SSP2-4.5 to be associated with a negative coefficient of such a small magnitude?

The range captures all possibilities, assuming each model is equally likely. This is determined by the change in IOD amplitude, which is somewhat similar to that of SSP1-2.6.

- It would be interesting in the discussion to provide a comparison with ENSO results to put in perspectives the magnitude of losses caused by IOD at a global and regional levels.

Good suggestion. We commented on global level as we don't have country-specific model for ENSO. For one standard deviation of the IOD time series, there is a loss of 0.24% cumulative loss in global economy. The 2019 strong pIOD event resulted in a loss of US\$558B, amounting to a reduction in the cumulative global GDP growth of 0.67%. By comparison, there is a ~1.5% cumulative loss in global economy per one standard deviation of the ENSO time series. The 2015/16 strong El Niño led to a loss of US\$3.9T, amounting to a ~5% cumulative loss in global economy. We have added this information. See Lines 161-163.

- Regarding climate variability and climate change effects on the global economy, it might be good to cite, in addition to references 31 and 32, the two articles by Kotz et al. (2021, 2022) in Nature Climate Change and Nature because they are recent significant contributions on this topic and address climate variability for temperatures and precipitation.

Done, thank you for the suggestion.

- The article's form could be improved. There are a few typos, such as 'Rsquare' instead of 'RSquared' as an example.

Done! Thank you.

Response to Reviewer #2

The authors provide a timely and interesting analysis of the impacts of the Indian Ocean Dipole on the economic growth of countries across the world. The paper builds on standard climate-econometric techniques but applies a nuanced understanding of climate science including EOF analysis and assessments of teleconnection strength to construct a powerful indicator of national-level exposure to IOD. The empirical results are interesting and the combination with future projections of IOD make for highly relevant findings. However, I have a number of questions regarding the methodology and framing which, for me, would be essential to address before publication. I list these in order of importance:

Thank you very much for your detailed, thoughtful, and helpful comments.

1. The independence of impacts from ENSO

Recent literature, including by the authors, has shown strong impacts of El Niño on economic growth. While the present analysis takes care to define the IOD teleconnections as independent of El Niño, they do not assess whether the economic impacts from IOD which they identify are independent from those caused by El Niño. Given that we know that positive IOD are often associated with El Niño, it is crucial for our understanding and the implications of the results to assess an empirical model which includes both an El Niño and an IOD index as independent variables to test how much they independently impact global economic output.

An IOD can occur independently of ENSO; in fact, most of the strongest positive IOD events (2019, 1961, and 1994) occurred in a non-El Niño year, and 2007 and 2008 positive IOD events occurred when the Pacific saw a La Niña event (Cai et al. 2013; Wang and Cai et al. 2024). It is true a positive IOD could be forced by an El Niño but an El Niño can also be forced by a positive IOD too. There is no obvious way to completely separate them, or their impact, because their mutual forcing/interaction is always present.

Our current work finds that for the IOD impact country-heterogeneity is essential; our Liu and Cai et al. paper finds that for ENSO a common shock dominates and that a nonlinear country-heterogeneous approach does not yield a statistically significant impact. We therefore test a new model that combines the two approaches with both an El Niño and an IOD index as independent variables, i.e., with IOD information removed from the ENSO index.

$$\begin{aligned} & \log(y_{i,t}) - \log(y_{i,t-1}) \\ &= \sum_{l=0}^n \{ \alpha_{1,l} \psi_l I_{t-l} + \alpha_{2,l} (\psi_l I_{t-l})^2 + \gamma_{1,l} E_{t-l} + \gamma_{2,l} E_{t-l}^2 + \beta_{1,l} T_{i,t-l} \\ & \quad + \beta_{2,l} T_{i,t-l}^2 + \lambda_{1,l} P_{i,t-l} + \lambda_{2,l} P_{i,t-l}^2 \} + \mu_i + \theta_{1,i} t + \theta_{2,i} t^2 + \varepsilon_{i,t} \end{aligned}$$

We find that the properties for each component, such as lag years and nonlinearity, are similar to their respective original models.

Compared with the respective original models, an in-dollar loss of individual events is reduced (**Fig. 1-R2**) because original ENSO and IOD “share a common loss” that is not reflected in their respective impacts.

The standard deviation of ENSO- and IOD-induced impact during 1960-2020 is 0.53% and 0.10%, respectively, indicating the ratio between IOD’s and ENSO’s impact is ~19%. This is similar to the estimate from the original respective models of ~16%.

Fig. 1-R2. | Similar to Fig. 1c of this paper and Fig. 2a-b of Liu et al. (2023). Shown are cumulative effects of a) two extreme pIOD and nIOD events and b) three extreme El Niño and La Niña events, but from the new model discussed above. The properties are similar.

References

Cai, W. *et al.* Projected response of the Indian Ocean Dipole to greenhouse warming. *Nature geoscience* **6**, 999-1007 (2013).

Wang, G. and Cai, W. *et al.* Change Indian Ocean Dipole in a warming climate. *Nature Review Earth and Environment*, in press (2014).

2. Projections of future impacts

The calculation of future economic impacts due to the IOD change should be made clearer, currently there are no equations for this in the methods section. In particular, the description on L241-246 was unclear to me in how the discount rate was applied. The discount rate should be used to weight the cumulative sum of future GDP losses, but in these lines it is described as being applied to the compounding growth rates. This method needs to be clarified before I can assess its validity.

Thank you for picking this up. Our description is **incorrect**. The discount rate was applied to weight the cumulative sum of future GDP losses. We have changed. See Lines 248-250.

Moreover, presenting economic damages as cumulative GDP losses over the whole of the century likely inflates the numbers compared to other presentations. Most assessments of future economic impacts of climate change typically report impacts as an annual % reduction of future GDP. Obviously, the authors are at liberty to choose how they present their results, but I think it would be valuable and make for a more honest comparison to at least also present the impacts of IOD that they calculate in this way which is consistent with other literature (e.g. Burke 2015, Kalkuhl & Wenz 2020).

Thank you for the suggestion. We have calculated the global GDP growth rate with and without IOD changes for the four emission scenarios. We average the growth rate across positive IOD events defined as when an SON IOD index exceeds a 1.0 s.d value (Fig. 2-R2). The increased intensity of pIOD in the future leads to an additional increase of 35%-47% per event in these scenarios (45% for SSP5-8.5).

Accumulating all events would lead to an even greater additional loss because the frequency also increases (Fig. 3-R2).

Fig. 2-R2 | Comparison of IOD-induced global GDP loss with and without IOD changes. a-d, pIOD-induced loss of global GDP growth with projected changing IOD (y-axis) and counterfactual IOD (x-axis), averaged in the period of 2020-2099 under the **a** SSP5-8.5, **b** SSP3-7.0, **c** SSP2-4.5, and **d** SSP1-2.6 scenario. A pIOD is defined as SON IOD index exceeds 1 s.d. Coloured dots indicate different climate models. Red pentagrams indicate the multi-model ensemble mean. The multimodel average for **a-d** (with, without) IOD changes are (-0.29%, -0.20%), (-0.28%, -0.19%), (-0.28%, -0.20%), and (-0.27%, -0.22%), for the **a** SSP5-8.5, **b** SSP3-7.0, **c** SSP2-4.5, and **d** SSP1-2.6 scenario, with a 45%, 47%, 40%, and 35% increase, respectively.

Fig. 3-R2 | Comparison of numbers of pIOD events with and without IOD changes. a-d, Same as Fig. 1R2, but for numbers of pIOD events with projected changing IOD (y-axis) and counterfactual IOD (x-axis), averaged in the period of 2020-2099 under the **a** SSP5-8.5, **b** SSP3-7.0, **c** SSP2-4.5, and **d** SSP1-2.6 scenario. The multimodel average for **a-d** (with, without) IOD changes are (23.4, 16.2), (22.4, 15.3), (20.4, 14.7), and (18.5, 16.3), for the **a** SSP5-8.5, **b** SSP3-7.0, **c** SSP2-4.5, and **d** SSP1-2.6 scenario, with a 44%, 46%, 39%, and 14% increase, respectively.

We have added the information in the revised version. See Lines 283-289. We have also included the two additional figures in the Supplementary Information (S11, S12).

3. Bootstrapping and significance of empirical regressions

The authors should include additional tables which show the results of the assessment of statistical significance when using the other two bootstrap resampling approaches. Moreover, the authors should discuss what assumptions and types of uncertainty these different bootstrapping approaches are likely to assess and justify their main choice. I think this is particularly important to consider in light of the fact that the main independent variable (IOD) is common, and therefore perfectly correlated, across all countries. This means that there are much fewer independent observations than normally assumed in a panel regression where independent variables are not so correlated across groups. This issue compared to normal panel models should be explicitly dealt with when presenting the significance of the econometric results.

It is true that the IOD is common to all countries; it is because the countries are commonly affected by the IOD that allows us to examine the impact. The interannual IOD index has little autocorrelation beyond lag 1 on both sides (0.15).

Thank you for the suggestion, we have now added additional Tables testing the statistical significance of different lists of years and 5-year blocks of countries (**Table 1-R2**, below). The Table is now included in the revised paper (**Supplementary Table S2**), with a comment in the main text, see Lines 134-137.

	Sampling by country	Sampling by year	Sampling by 5-year block
$\alpha_{1,0}$	-0.0016* (5.93×10^{-4})	-0.0015* (7.24×10^{-4})	-0.0016* (6.35×10^{-4})
$\alpha_{1,1}$	-0.0018* (6.69×10^{-4})	-0.0017* (8.17×10^{-4})	-0.0018* (8.78×10^{-4})
$\alpha_{1,2}$	-0.0017* (6.82×10^{-4})	-0.0017* (7.74×10^{-4})	-0.0017* (7.65×10^{-4})
$\alpha_{1,3}$	-7.24×10^{-4} (7.59×10^{-4})	-6.94×10^{-4} (7.90×10^{-4})	-7.86×10^{-4} (5.90×10^{-4})
$\alpha_{2,0}$	4.05×10^{-4} (2.57×10^{-4})	3.51×10^{-4} (1.64×10^{-4})	3.45×10^{-4} (1.76×10^{-4})
$\alpha_{2,1}$	-7.61×10^{-4} *** (1.74×10^{-4})	-7.49×10^{-4} *** (2.61×10^{-4})	-7.37×10^{-4} *** (3.09×10^{-4})
$\alpha_{2,2}$	-2.41×10^{-5} (2.25×10^{-4})	-2.42×10^{-5} (2.12×10^{-4})	-3.21×10^{-5} (1.93×10^{-4})
$\alpha_{2,3}$	-2.50×10^{-4} (2.28×10^{-4})	-2.56×10^{-4} (1.43×10^{-4})	-2.56×10^{-4} (1.15×10^{-4})

Table 1-R2 | Regression coefficients α from three bootstrap resampling approaches in various lag years. Values in the brackets denote s.d. of 1,000 bootstrapped coefficients. Superscript *, ** and *** indicate the estimate of coefficient is statistically significant above the 90%, 95% and 99% confidence level, respectively.

4. Climate teleconnection

This is an interesting and subtle way to construct a countries potential exposure to IOD. However, a few methodological choices seem subjective and alternative approaches are not explored. E.g.: Can the authors demonstrate the effectiveness of the partial regression by also showing the direct correlation of sea surface temperatures with IOD without removing the ENSO effect?

We have added. Without removal of ENSO, one sees strong ENSO anomalies (see **Fig. 4-R2** below).

Fig. 4-R2 | Illustration of effectiveness of a partial regression. a-h, Monthly regression coefficients of observed sea surface temperature onto the observed SON IOD index, from May to December month by month. Only the regression coefficients above the 95% confidence level are shown. Without removing ENSO, ENSO signal is clearly seen in the Pacific.

Why are climate teleconnections only from May to December included?

This is because an IOD develops in June, but ends by November, and no IOD survives beyond December. This is determined by strong seasonal cycle in the Indian Ocean. We have added this information to the paper. See Lines 102-103.

Minor:

Fixed-effects choices.

Panel fixed-effects models typically include two way (country and year) fixed effects for causal identification. The year fixed effects capture unobserved confounders which are common across countries in given years. Here, these are not included. This is likely due to the fact that IOD variation is common across countries and so including year-fixed effects would remove the variance in which the model is interested in identifying. So this is a justifiable choice, but the authors should make that justification explicit.

We thank the reviewer for raising this point. As you pointed out, the IOD is a time-specific global phenomenon, like one of other major events (e.g., financial crisis), that commonly affects countries. Including year-fixed effects weakens the statistical influence of the IOD, leading to an underestimation of the real impact of the IOD on economic growth, as pointed out previously (Dell et al. 2015). Also, year-fixed effect could introduce risk of collinearity as IOD time-fixed effects could be correlated with time-specific factors, making it harder to separate impacts from the IOD. We have made these clear, see Lines 366-369.

Description as “country-specific” results

In my opinion, the description of the empirical model as “country-specific” is mis-leading. A global panel model estimates an aggregated response across countries. Even when including interaction terms (here with the teleconnection strength) to describe cross-country heterogeneity, the model still estimates a global heterogeneity based on this strength rather than a specific country-level response.

We have changed to “country-heterogenous” throughout the paper.

L144 – do the authors distinguish between countries with a larger/smaller agricultural share in their econometric specification? It appears not based on their methods, and that the only way different impacts across countries are described in the model are through the teleconnection strength. Therefore, it seems like this interpretation of their results that Australia suffers a larger impact due to its larger agricultural sector is not justified by their methods or results.

We have removed the inference, see Lines 146-148.

L133 – can the authors show a table indicating this insignificant result?

We are sorry, our statement there was incorrect; we meant a nonmeaningful result. Some coefficients are significant but for year 0 (occurrence year) both a positive and a negative IOD produce a benefit due to a positive $\alpha_{2,0}$. We have added this information to the paper, see Lines 385-388.

140 - the impact is of an opposite sign for negative/positive IOD and this should be stated, not just that the impact is greater for positive than negative.

Done. Thank you for the suggestion. We have added a comparison. For example, a +1.0 s.d. pIOD causes -1.89% ($\pm 0.64\%$) loss in Kenya GDP growth that is statistically significant, while a -1.0 s.d. nIOD causes +1.09% ($\pm 1.25\%$) benefit in Kenya GDP growth that is not statistically significant. See Lines 142-145.

L207-210 – should be made clearer that these sentences refer to the teleconnection strengths/patterns.

Great point. Made clear now. See Lines 214-215.

There are a number of minor wording issues which should be corrected, in particular missing “the”, e.g. L87 “the impact on the global economy”.

Thanks, changed in appropriate places.

Response to Reviewer #3

This paper estimates the effect of the Indian Ocean Dipole (IOD) on economic growth, and projects the economic impacts of an intensified IOD amplitude under future climate change. The IOD is a mode of interannual global climate variability originating in sea surface temperature anomalies in the equatorial Indian Ocean. These anomalies are correlated with strong variation in surface air temperature and rainfall in some parts of the world (i.e., highly “teleconnected”) but much less so in other parts of the world (i.e., not highly “teleconnected”). The paper exploits these differences in teleconnection to estimate the effect of the IOD on economic growth by country. An important finding is that the effects are highly nonlinear/assymmetric- while a large positive IOD has a large negative effect on economic growth, a negative IOD has virtually no effect on growth. The harmful effects of positive IODs are stronger for more teleconnected countries. Based on these findings, the paper projects that intensified IOD amplitude under climate change will have large economic costs compared to a no-climate-change counterfactual. These costs will be disproportionately borne by today’s developing and emerging economies, as these happen to be the ones with greatest teleconnection. (Climate change is not projected to alter the patterns of teleconnection.)

The basic research design of this paper combines previous work on the effects of another mode of interannual global climate variability- ENSO (e.g., Hsiang et al., 2011) with previous work on the effects of climate on economic growth (e.g. Burke et al., 2015). ENSO originates in sea surface temperature anomalies in the eastern equatorial Pacific and its effects have been studied considerably more than those of the IOD. As such, this paper fills in important gap by studying the economic effects of the IOD.

While the paper takes on an important topic, more work needs to be done on the econometric estimation and climate change impact projections to ensure the robustness of the results. The main estimating equation (Equation 2) includes a multiplicative interaction of an annual IOD index value with a countryspecific measure of teleconnection. This interaction term is modeled as a quadratic function to account for nonlinearity in the country-specific effects of IOD. Additional regressors include country-level annual mean surface air temperature and rainfall.

Thank you for the interest and positive comments on our attempt.

1. Importantly, the IOD index values used in the estimating equation are constructed via a partial regression that removes the impact of ENSO (Equation 1). This approach seems opaque. The authors should consider also running an alternative version of the estimating equation that includes as regressors both the “IOD index x IOD teleconnection” interaction as well as an ENSO index x ENSO teleconnection interaction. Such a specification will directly control for the effect of ENSO and will allow for a transparent comparison between the effects of IOD and ENSO. Both effects could then also be carried through to the climate change impact projections, allowing for a comparison of the respective impacts.

We thank the reviewer for the vision of such a unified model assessment. We build a model of country-specific IOD impact and country-specific ENSO impact as below:

$$\log(y_{i,t}) - \log(y_{i,t-1}) = \sum_{l=0}^n \{ \alpha_{1,l} \psi_l I_{t-l} + \alpha_{2,l} (\psi_l I_{t-l})^2 + \gamma_{1,l} \varphi_l \mathbf{E}_{t-l} + \gamma_{2,l} (\varphi_l \mathbf{E}_{t-l})^2 + \beta_{1,l} T_{i,t-l} + \beta_{2,l} T_{i,t-l}^2 + \lambda_{1,l} P_{i,t-l} + \lambda_{2,l} P_{i,t-l}^2 \} + \mu_i + \theta_{1,i} t + \theta_{2,i} t^2 + \varepsilon_{i,t},$$

where ENSO teleconnection is obtained as in Liu et al. (2023). As can be seen from the regression coefficients shown in **Table 1-R3**, there is little significance of a country-specific ENSO impact. Our work on ENSO impact (Liu et al. 2023) found that a teleconnection-

weighted approach produces nonlinear (Alpha1) and nonlinear (Alpha2) coefficients that are mostly NOT statistically significant, suggesting that ENSO is mostly consequential in terms of its cascading effect and its common spill-over to the global economy. We have included this information, see Lines 388-390.

However, the “mother” board of our current model and that in Liu et al 2023 is the same (based on Burke et al. 2015) and in this sense because we have made the IOD independent from ENSO, our results here and in Liu et al. 2023 are by and large addable. Globally, impact from the IOD is about ~16.0% of that from ENSO. We have added discussion to this effect; see Lines 391-392.

IOD x IOD teleconnection with ENSO x ENSO teleconnection			
$\gamma_{1,0}$	-0.0016** (6.35×10^{-4})	$\gamma_{2,0}$	-1.37×10^{-4} (1.08×10^{-4})
$\gamma_{1,1}$	-5.19×10^{-4} (6.50×10^{-4})	$\gamma_{2,1}$	2.02×10^{-4} * (1.10×10^{-4})
$\gamma_{1,2}$	-9.43×10^{-4} (6.51×10^{-4})	$\gamma_{2,2}$	-6.54×10^{-5} (1.18×10^{-4})
$\gamma_{1,3}$	-5.60×10^{-4} (6.02×10^{-4})	$\gamma_{2,3}$	-4.69×10^{-5} (1.19×10^{-4})

Table 1-R3 | Coefficients of ENSO x ENSO teleconnection interaction.

2. The estimating equation includes country-level annual mean surface air temperature and rainfall, both modeled as quadratic. However, the estimated effects of these variables on economic growth are never displayed, nor do they seem to be accounted for in the climate change impact projections. Considering that prior work has shown temperature (Burke et al., 2015) and rainfall (Kotz et al., 2022) to affect economic growth, the authors should present the estimated effects of these variables and carry these forward when projecting the impacts of climate change. This will allow for comparison between the impacts of different climatic variables on future economic costs.

Again, this is a grand vision which we would aspire to eventually have. The impact on global economy by changing temperature and rainfall has been studied substantially; that is why we have not focused. Our model reveals a similar nonlinear simultaneous effect of temperature with Burke et al. (2015 & 2018), of which both linear and quadratic coefficients are statistically significant ($\beta_1 = 0.0107$, standard error 0.0031, $p < 0.001$; $\beta_2 = -3.36 \times 10^{-4}$, standard error 9.66×10^{-5} , $p < 0.001$). The slight difference between our estimation and Burke et al (2015) is probably due to: (1) we remove IOD’s signal in the annual temperature; (2) we exclude the year-fixed effect to avoid collinearity as IOD time-fixed effects could be correlated with time-specific factors. Projecting the economic impact from changes in temperature and rainfall is not what we are set out to obtain. We are focusing on economic impact of the IOD, and its future changes, assuming that the functional link in terms of Alpha1 and Alpha2 remains the same in the future. We have added discussion on this; see Lines 367-371.

3. The estimating equation includes contemporaneous and lagged effects of all variables. However, only the cumulated effect is ever plotted. It would be useful to also see the separated effects presented in a supplementary figure.

Thank you for the suggestion. We have now included this information (see Fig. 1-R3 below) as **Supplementary Fig. 4**. The call out is at Line 132.

Fig. 1-R3 | Same as Fig.1c. but for IOD-induced global GDP change at each lag year.

Other Points

Figure 2c plots the nonlinearity of economic impact (referred to as alpha) against economic teleconnection strength. It is unclear to me what exactly is alpha as I could not find it defined anywhere. The estimating equation (Equation 2) refers to alpha_1 and alpha_2, but not alpha without a subscript.

Sorry for the confusion. Alpha is completely different from Alpha1 and Alpha2. We have change Alpha to Omega.

2. There are multiple references to an acronym “SON”, but I could not find it defined anywhere.

Defined. It is September, October, and November. See Line 42. Thanks.

References

The three papers are cited.

Burke, M., Hsiang, S.M. and Miguel, E., 2015. Global non-linear effect of temperature on economic production. *Nature*, 527(7577), pp.235-239.

Hsiang, S.M., Meng, K.C. and Cane, M.A., 2011. Civil conflicts are associated with the global climate. *Nature*, 476(7361), pp.438-441.

Kotz, M., Levermann, A. and Wenz, L., 2022. The effect of rainfall changes on economic production. *Nature*, 601(7892), pp.223-227

Reviewers' Comments:

Reviewer #1:

Remarks to the Author:

I thank the authors of the article for their responses and the quality of the discussions. I am pleased to see that several of my suggestions have further improved the article. This article now should be published in Nature Comm'.

Here are my feedback points that may not necessarily require a response:

You mentioned that the NL model explains approximately 50% more variability than that explained by the linear model; it would have been interesting to provide the adjusted R-squared and the sum of squares explained elsewhere. Please correct in the text that this refers to the adjusted R-squared, which is still not the case, and that the choice of lags in the ARDL model was made using this indicator.

It is interesting to learn that the common shock of IOD is not meaningful and represents only 16% of that of ENSO. Similarly, the indirect effect captured by an interaction term like $T*IOD$ is not significant. That being said, perhaps the effect of climate change would be more visible by considering ΔT ...

Thank you for the new Figure 1, which now incorporates the USA and the UK; I believe this is informative.

Lastly, thank you for conducting robustness tests, particularly by considering sub-periods. The fact that the linear coefficient no longer emerges significantly (although it remains stable in magnitude) over the most recent period can indeed be explained by the reduction in sample size, and we may be capturing less of the long-term average effect. However, wouldn't bootstrapping help compensate for this? This aligns with the a priori absence of the effect of the interaction term $T*IOD$: the acceleration of global warming would not therefore modify the effect of IOD on economic growth.

Reviewer #2:

Remarks to the Author:

I thank the authors for their detailed responses to my comments and suggestions which have satisfied a number of my concerns.

One thing still remain which I believe the authors could present better to communicate the robustness of their results.

1. Independence of ENSO and IOD

In their response to my first point in the previous round of reviews, the authors conducted an additional analysis which showed that while the effect of IOD on the economy is largely independent of that of ENSO, some of the effect is shared and hence the independent effect of IOD is marginally smaller (19%). I think it is in the interest of scientific integrity (and would as such actually support the main manuscript) for the authors to report the additional figure they shared in their response letter in the supplementary materials and to comment on this inter-dependence between ENSO and IOD impacts in the main manuscript. The authors should not worry, as this will not undermine their results but strengthen the reader's confidence in the manuscript and ability to infer the broader picture from the results.

I thank the authors for addressing all my other points in a very good manner! Best wishes to them for their future research. Max Kotz.

Reviewer #3:

Remarks to the Author:

Thank you for addressing my comments from the first review. I find the paper is considerably stronger now.

However, I continue to be concerned about the partial, incomplete nature of the projection exercise, which only projects the impacts of future IOD changes but not temperature and rainfall changes, even though these are estimated in the model (Equation 2).

As I understand, it is feasible to project the full impact of changes in climatic variables implied by the empirical model, though please correct me if I am wrong. The projected impacts of IOD changes will not be altered when doing this; they will simply be put into larger perspective, and the projection exercise will be more transparent.

Response to Reviewer #1

I thank the authors of the article for their responses and the quality of the discussions. I am pleased to see that several of my suggestions have further improved the article. This article now should be published in Nature Comm'.

Thank you!

Here are my feedback points that may not necessarily require a response:

You mentioned that the NL model explains approximately 50% more variability than that explained by the linear model; it would have been interesting to provide the adjusted R-squared and the sum of squares explained elsewhere. Please correct in the text that this refers to the adjusted R-squared, which is still not the case, and that the choice of lags in the ARDL model was made using this indicator. It is interesting to learn that the common shock of IOD is not meaningful and represents only 16% of that of ENSO. Similarly, the indirect effect captured by an interaction term like $T \cdot IOD$ is not significant. That being said, perhaps the effect of climate change would be more visible by considering ΔT ...

Adjusted R-squared is mentioned. The "T" here means ΔT .

Thank you for the new Figure 1, which now incorporates the USA and the UK; I believe this is informative. Lastly, thank you for conducting robustness tests, particularly by considering sub-periods. The fact that the linear coefficient no longer emerges significantly (although it remains stable in magnitude) over the most recent period can indeed be explained by the reduction in sample size, and we may be capturing less of the long-term average effect. However, wouldn't bootstrapping help compensate for this? This aligns with the a priori absence of the effect of the interaction term $T \cdot IOD$: the acceleration of global warming would not therefore modify the effect of IOD on economic growth.

Bootstrapping only helps compensate for this mildly.

Reviewer #1 (Remarks on code availability):

Very small code written for Matlab. Not easy if you have not access to Matlab.

Right, that is the code we developed.

Response to Reviewer #2

I thank the authors for their detailed responses to my comments and suggestions which have satisfied a number of my concerns.

One thing still remain which I believe the authors could present better to communicate the robustness of their results.

1. Independence of ENSO and IOD

In their response to my first point in the previous round of reviews, the authors conducted an additional analysis which showed that while the effect of IOD on the economy is largely independent of that of ENSO, some of the effect is shared and hence the independent effect of IOD is marginally smaller (19%). I think it is in the interest of scientific integrity (and would as such actually support the main manuscript) for the authors to report the additional figure they shared in their response letter in the supplementary materials and to comment on this inter-dependence between ENSO and IOD impacts in the main manuscript. The authors should not worry, as this will not undermine their results but strengthen the reader's confidence in the manuscript and ability to infer the broader picture from the results. I thank the authors for addressing all my other points in a very good manner! Best wishes to them for their future research. Max Kotz.

Thank you Dr Kotz, we have added an additional figure as you suggested.

Response to Reviewer #3

Thank you for addressing my comments from the first review. I find the paper is considerably stronger now. However, I continue to be concerned about the partial, incomplete nature of the projection exercise, which only projects the impacts of future IOD changes but not temperature and rainfall changes, even though these are estimated in the model (Equation 2).

As I understand, it is feasible to project the full impact of changes in climatic variables implied by the empirical model, though please correct me if I am wrong. The projected impacts of IOD changes will not be altered when doing this; they will simply be put into larger perspective, and the projection exercise will be more transparent.

Our study aims to investigate the impact from changing IOD. We avoid repeating work of previous studies.